# Sim2Reason: Solving Physics Olympiad via Reinforcement Learning on Physics Simulators

**Mihir Prabhudesai** [* † 1] **Aryan Satpathy** [* † 1] **Yangmin Li** [† 1] **Zheyang Qin** [† 1]
**Nikash Bhardwaj** [1] **Amir Zadeh** [2] **Chuan Li** [2] **Katerina Fragkiadaki** [1] **Deepak Pathak** [1]

## Abstract

We have witnessed remarkable advances in LLM reasoning capabilities with the advent of DeepSeek-R1. However, much of this progress has been fueled by the abundance of internet question–answer (QA) pairs—a major bottleneck going forward, since such data is limited in scale and concentrated mainly in domains like mathematics. In contrast, other sciences such as physics lack large-scale QA datasets to effectively train reasoning-capable models. In this work, we show that physics simulators can serve as a powerful alternative source of supervision for training LLMs for physical reasoning. We generate random scenes in physics engines, create synthetic question–answer pairs from simulated interactions, and train LLMs using reinforcement learning on this synthetic data. Our models exhibit zero-shot sim-to-real transfer to real-world physics benchmarks: for example, training solely on synthetic simulated data improves performance on IPhO (International Physics Olympiad) problems by 5-10 percentage points across model sizes. These results demonstrate that physics simulators can act as scalable data generators, enabling LLMs to acquire deep physical reasoning skills beyond the limitations of internet-scale QA data. Code available at: https://sim2reason.github.io/.

## 1. Introduction

Reinforcement learning with verifiable rewards (RLVR) has enabled large language models (LLMs) to cross the threshold from pattern matching to multi-step reasoning. However, this progress is limited by the availability of high-quality question–answer (QA) pairs: textbook and internet-derived QA corpora are finite, unevenly distributed across domains, and difficult to scale beyond a few million examples. As a result, RLVR systems such as DeepSeek-R1 (DeepSeek-AI et al., 2025) are bottlenecked not by model capacity, but by scarcity of supervision data (Wu et al., 2025).

This limitation is most visible in the physical sciences. While mathematics benefits from abundant question–answer pairs, physics, chemistry, and other empirical sciences lack comparable large-scale datasets. For example, less than 1% of the 800K QA pairs used in DeepSeek-R1 involve STEM topics, leading to poor generalization on standard physics benchmarks. The root issue is that internet QA data is sparse, unevenly distributed, leaving large gaps in the supervision signal required for scientific reasoning.

Physics engines, on the other hand, encode physical laws in executable form. Instead of describing phenomena in text, they compute future states by numerically integrating systems of ordinary differential equations under constraints. This gives them the ability to generate unlimited trajectories with high-fidelity supervision signals—such as instantaneous forces, momentum, and energy transfers—that are rarely captured in static internet corpora. However, this information is not directly usable by LLMs to improve their physics problem solving skills: simulator outputs are approximate, continuous, forward-time numerical traces, whereas physics problem solving requires accurate, inverse, symbolic, and counterfactual reasoning. The challenge, then, is how to represent simulator-derived physical information in a way that helps improve an LLM's physics problem solving ability.

One potential solution is utilizing physics simulators as external tools (Schick et al., 2023; Sarch et al., 2025). However, this approach is non-trivial as it shifts the primary challenge from physical reasoning to code generation; the LLM must master complex simulator-specific APIs to model a problem. Our early experiments with this paradigm were unsuccessful, as models frequently struggled to produce executable and physically accurate simulation code. Further-

---

* Co-lead and equal contribution † Core contributor [1]Carnegie Mellon University [2]Lambda. Correspondence to: Mihir Prabhudesai <mprabhud@andrew.cmu.edu>, Aryan Satpathy <asatpath@andrew.cmu.edu>.

*Proceedings of the $43^{rd}$ International Conference on Machine Learning*, Seoul, South Korea. PMLR 306, 2026. Copyright 2026 by the author(s).

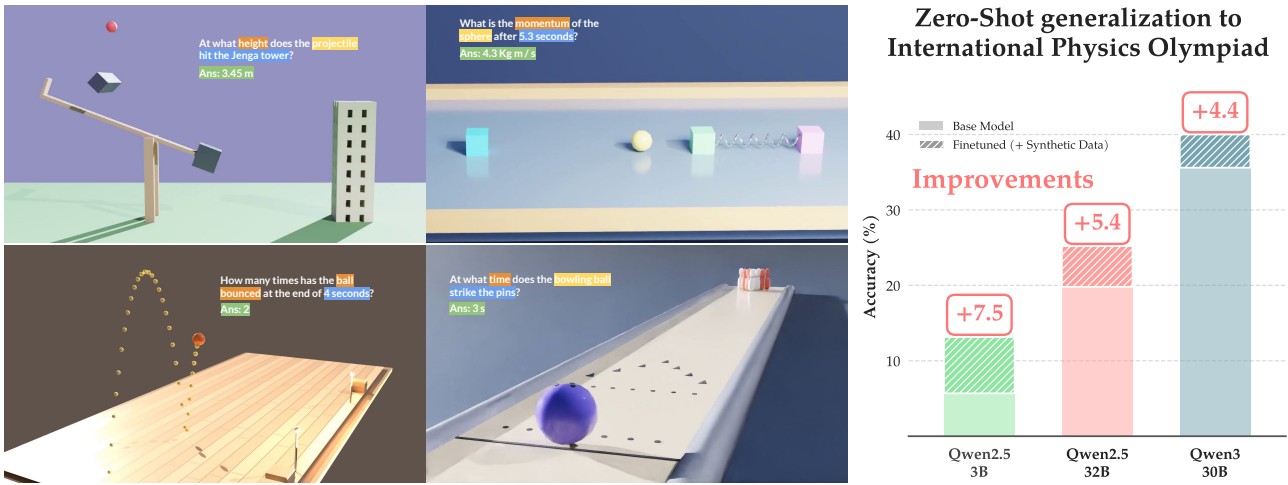

*Figure 1.* We present **SIM2REASON**: a method for turning physics simulators into scalable generators of question–answer pairs to improve LLM reasoning, removing the need of human annotation in the data-generation pipeline. The core idea is to **structure the randomization with a domain-specific language (DSL)** and use it to procedurally generate reasoning problems, as illustrated in the examples above. LLMs finetuned on this synthetic data get **zero-shot improvement on real world benchmarks** such as International Physics Olympiad.

more, many physical phenomena are not natively supported by simulators, and implementing them requires human-in-the-loop engineering, which renders this approach unscalable. In contrast, we find that our method allows us to generalize beyond the scope of our simulator (Section 3.5).

To address these limitations, we propose Sim2Reason: a framework that transforms the physics simulator into a scalable QA generator (Figure 1). Instead of relying on the LLM's initial coding capabilities, we procedurally construct diverse physical systems in the physics simulator and simulate their dynamics to automatically generate verified question-answer pairs. Our pipeline produces three reasoning modes: numeric (state queries), reverse (parameter inference), and symbolic (closed-form expressions). These systems span a broad spectrum of classical mechanics, covering the majority of core phenomena encountered in undergraduate and Olympiad-level physics. The procedural nature of our Domain Specific Language (DSL) enables the dynamic composition of heterogeneous physical scenes—such as combining pulley systems with rotational dynamics—generating millions of unique, physically grounded training samples (Figure 2).

We train LLMs using Reinforcement Learning (RL) on this synthetic data without incorporating any real-world physics QA pairs during the post-training phase. Evaluating our model across multiple rigorous benchmarks—including IPhO, JEE-Bench, PHYSICS and OlympiadBench—reveals consistent and meaningful performance gains, showcasing a robust sim-to-real transfer. We find that quality filtrating is critical to achieving these gains. For instance, simulator-generated questions often suffer from degeneracy, where

problems are either trivially easy or computationally intractable. To address this, we implement a question pruning strategy that filters out these extremes, ensuring training compute is focused on useful samples that fall within the LLM's solvable range.

Our results demonstrate that training solely on Sim2Reason data improves zero-shot performance on IPhO mechanics problems by 5–10 percentage points across 3B to 32B model scales. We observe similar gains on specialized benchmarks like JEEBench (+17.9% for 32B models) and PHYSICS, confirming that the model is not merely memorizing simulator dynamics but is developing a generalized capacity for multi-step physical reasoning. Furthermore, we find that the QA pairs generated by our framework serve as an effective benchmarking tool for foundation models. We observe a high correlation between model accuracy on our simulated questions and performance on real-world physics benchmarks, enabling scalable and automated testing across specific physical domains. Please refer to our project webpage, for code and video visualizations from SIM2REASON: https://sim2reason.github.io/

## 2. Method

To train LLMs for physical reasoning, we first generate synthetic data using a physics simulator and then fine-tune the LLM on this synthetic data. Using MuJoCo (Todorov et al., 2012) as our simulator, we generate QA pairs spanning a wide range of physical phenomena, broadly covering kinematics, rotational mechanics, orbital motion, variable-mass systems, and basic electromagnetism (e.g., a charged particle moving in the presence of time-varying fields).

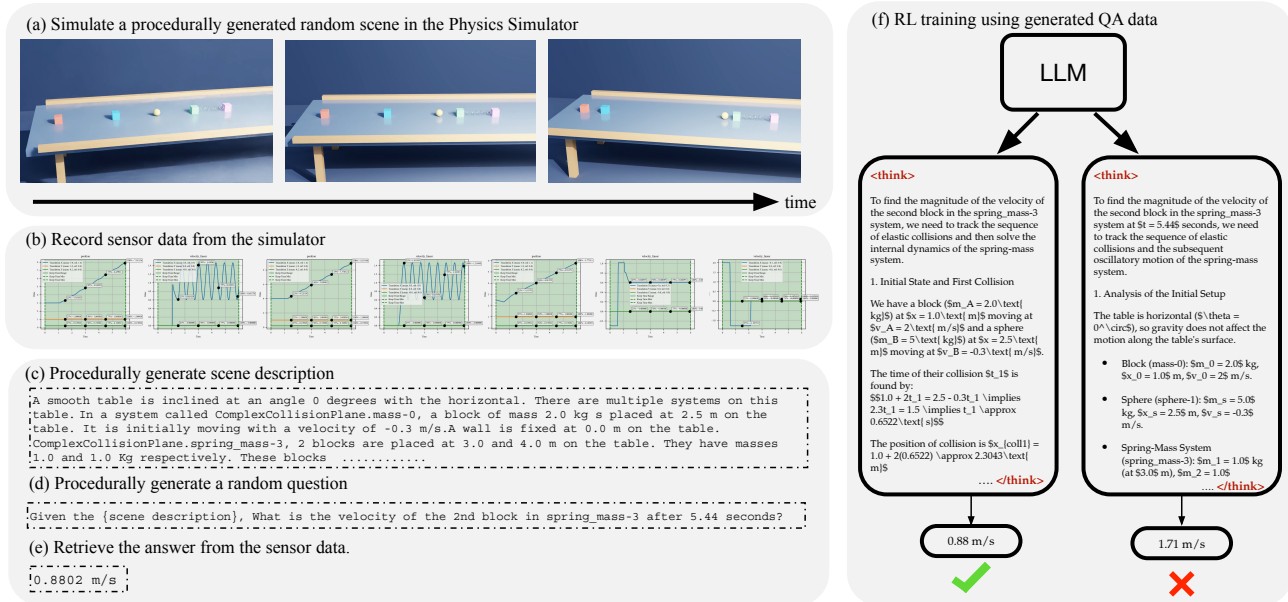

*Figure 2.* Overview of the SIM2REASON (Sim2Reason) pipeline. From left to right: we procedurally generate diverse physics scenes using a DSL, (a) compile them into MuJoCo simulations, and (b) record physically grounded state/force traces. (c–e) From these traces we automatically instantiate multiple types of question–answer pairs (numeric, reverse, and symbolic), and apply filtering to remove degenerate/shortcut questions and unstable simulation segments. (f) Finally, we post-train an LLM with RLVR on the resulting synthetic data and evaluate zero-shot sim-to-real transfer on real-world benchmarks (e.g., IPhO and other physics/math datasets).

The data generation pipeline (Figure 3) consists of 4 stages:

1. **Scene Generation**: Generating physically meaningful random scenes

2. **Physics Simulation**: Simulating scenes to record data

3. **QA pair generation**: Generating question-answer pairs from recorded data

4. **Data filtration**: Deduplicating and filtering degenerate qa pairs

### 2.1. Scene Generation

To procedurally generate scenes in a structured and scalable manner, we design a domain-specific language (DSL) that isolates physically meaningful axes of randomization from those that do not fundamentally change the underlying reasoning (Figure 7). For example, changing the length of a pulley string typically does not affect the system's dynamics, whereas changing the mass of a suspended block does.

Our DSL consists of three levels of abstraction: scene, entity, and body. **Body** is the most fundamental element. Each body has a name and a predefined set of parameters based on its type—for instance, the mass of a block or the radius of a sphere (see Appendix D for details). Additionally, for each body we define a template MuJoCo XML snippet and a template string that describes the body and its parameters.

However, bodies cannot be arbitrarily connected—for instance, a mass block can be placed on a prism, but not vice versa. This motivates the next level of abstraction: an **entity**, which consists of a set of bodies connected in a specific, physically meaningful way. Each entity exposes well-defined connection points that specify how it can attach to other entities. We refer to Appendix F for a detailed list of entities.

The **scene** is formed by randomly selecting entities and connecting them. We generate the MuJoCo XML for a scene by concatenating the XML templates of its entities, each of which is in turn constructed by composing the XML templates of its bodies. This design allows us to generate simulatable scenes at scale without a human in the loop (Figure 7 in Appendix).

### 2.2. Physics Simulation

To generate synthetic data, we simulate the generated scenes in MuJoCo and record key physical quantities for each body. We categorize bodies into either masses (proprioceptive quantities) or strings (tension and length); Appendix E lists all recorded quantities.

However, the recorded traces can contain unmodeled transitions—such as a block colliding with a pulley or falling off a plane—that lead to unpredictable dynamics. We detect these events by comparing the sliding-window mean and

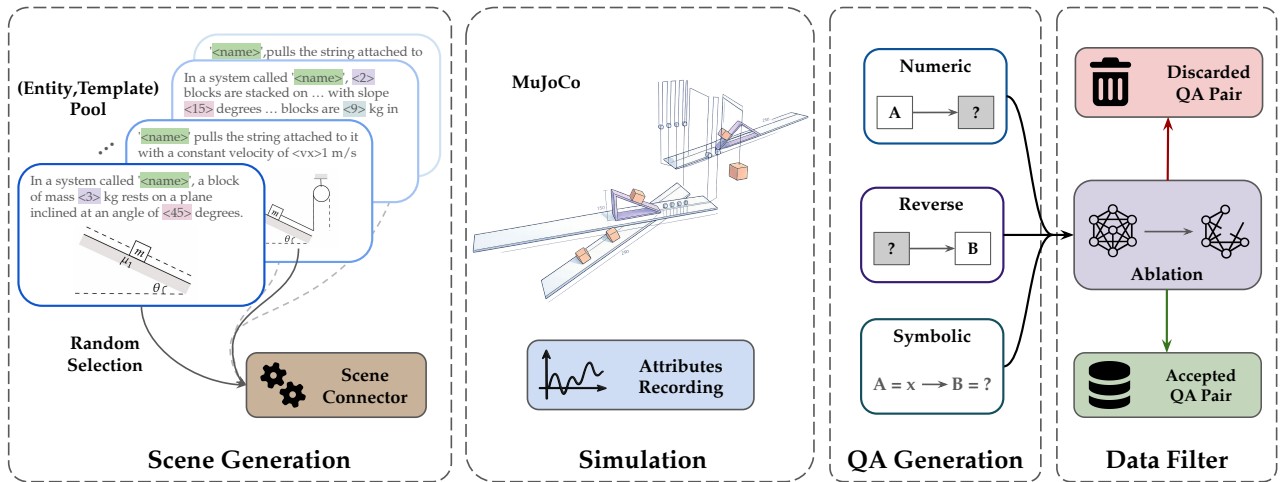

*Figure 3.* Overview of our synthetic data-generation pipeline. We procedurally generate simulatable scenes by randomly selecting and connecting DSL entities (Section 2.1), then simulate each scene in MuJoCo and record time-series data of key physical attributes (Section 2.2). From these traces we craft natural-language QA pairs in three formats (Section 2.3)-numeric, reverse, symbolic-and finally deduplicate and filter degenerate/shortcut-solvable questions before RL post-training (Section 2.4).

standard deviation. More specifically,

$$
\begin{aligned}
\mu_t &= \mathrm{mean}\{a_j\}_{j=t}^{t+w}, \\
\sigma_t &= \mathrm{std}\{a_j\}_{j=t}^{t+w}, \\
\text{truncate at } t \text{ if } \max_{i\in\{t,\dots,t+w\}} &|a_i - \mu_t| \geq k\,\sigma_t.
\end{aligned} \tag{1}
$$

Here, $a$ denotes the recorded acceleration of a body, and $k$ is a threshold hyperparameter controlling how aggressively we flag spikes (smaller $k$ is more sensitive to spikes). We use $k = 5$ during data generation.

An example of this pruning procedure is shown in Figure 8 in Appendix. We also extend the simulator to support variable-mass systems, Newtonian gravitation, and collisions with a specified coefficient of restitution.

### 2.3. QA Pair Generation

For a given simulatable scene, we convert its recorded time-series data into natural-language question–answer pairs. We first generate a scene description by concatenating the natural-language descriptions of its entities (themselves composed from body descriptions). We also describe inter-entity connections using reusable template strings for each connection mode.

To form a question, we randomly select a body, a recorded physical quantity, and a timestep. We generate questions in three ways, each requiring a different style of reasoning:

- **Numeric questions:** Forward reasoning, e.g., "What

is the velocity of block A at time 3 s?"

- **Reverse questions:** Inverse reasoning, where one scene parameter is masked (e.g., $x$), e.g., "What is the mass of block A if its velocity after 3 s is 5 m/s?"

- **Symbolic questions:** Symbolic reasoning, where all numeric parameters are replaced by symbols, e.g., "What is the velocity of block A after time $t$?"

### 2.4. Data Filtration

We filter the generated data to remove *shortcut solutions*, i.e., cases where a model can ignore part of the scene (or collapse a multi-body interaction into an oversimplified system) and still obtain the correct numeric answer (Figure 4). This is undesirable for RL training because it can reward incorrect physical reasoning and reinforce approximations.

To detect shortcut-solvable questions, we construct controlled "ablations" of each scene:

- **Entity-removal ablations:** We treat a scene as a graph of entities and connections, generate sub-scenes by removing one entity at a time while preserving the connectivity of the remaining graph, and re-simulate these sub-scenes.

- **Joint-removal ablations:** We generate variants in which individual joints/constraints are replaced by rigid "glued" components.

For a given question, if the ground-truth answer is unchanged between the original scene and *any* ablated variant,

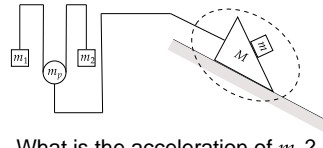

What is the acceleration of $m_1$?

*Figure 4.* Illustration of a *shortcut solution*. The correct answer depends on the coupled motion of the block $m$ and wedge $M$, but weaker models may collapse the dotted region into a single body of mass $M + m$ and still match the numeric answer. We filter QA pairs whose answers are invariant to such approximations.

we discard the QA pair. This prunes questions whose solution does not actually depend on the purported multi-entity dynamics and can be solved by approximating the scene with an oversimplified setup. In practice, approximately 15% of generated QA pairs are filtered out by this procedure.

## 2.5. RL Training

We post-train the LLM using reinforcement learning with verifiable rewards (RLVR). For each prompt $x$, we sample a group of $G$ responses $\{y_i\}_{i=1}^G$ from the current policy $\pi_\theta(\cdot \mid x)$ and assign a scalar reward $R(x, y_i)$ based on final-answer correctness. We assign a positive reward when the model's final answer lies within a 5% relative error of the simulator-recorded value; otherwise, the reward is zero. This tolerance accounts for the numerical approximations in the simulator, while penalizing incorrect physical reasoning. We optimize Group Sequence Policy Optimization (GSPO)(Zheng et al., 2025a) with a reference policy $\pi_{\text{ref}}$ (the base Instruct model).

As is common in group-based RL, we compute group-relative advantages by normalizing rewards within each group (subtracting the group mean and dividing by the group standard deviation). The GSPO loss is a clipped, sequence-level policy-gradient objective:

$$\mathcal{L}_{\text{GSPO}}(\theta) = -\mathbb{E}_{x,\{y_i\}} \left[ \frac{1}{G} \sum_{i=1}^G \min\Big( \rho_i \hat{A}_i, \right.$$
$$\left. \text{clip}(\rho_i, 1 - \varepsilon, 1 + \varepsilon) \hat{A}_i \Big) \right] \qquad (2)$$

where $\rho_i = \pi_\theta(y_i \mid x) / \pi_{\text{old}}(y_i \mid x)$.

Finally, we incorporate DAPO-style *dynamic sampling* to improve training efficiency in sparse-reward settings. Concretely, if a sampled prompt yields near-zero reward standard deviation across the group (leading to near-zero advantages), we resample additional prompts until the batch is filled with informative groups.(Yu et al., 2025)

## 3. Experiments

We evaluate our proposed SIM2REASON pipeline by post-training LLMs of various sizes with reinforcement learning (RL) on our synthetic dataset. We then test these resulting models on real-world reasoning benchmarks.

**Datasets & Evaluation:** Below we describe the datasets we use for training and evaluation.

- **Synthetic (SIM2REASON):** We generate training questions on-the-fly using the proposed SIM2REASON pipeline; unless stated otherwise, all RL runs use this synthetic distribution. We use numeric QA mode as described in Section 2.3, for all our training runs, we compare against symbolic and reverse QA mode in our ablation section.

  Concretely, we train for 200 RL steps with batch size 32, so the model observes approximately 6,400 distinct question–answer pairs during post-training.

- **International Physics Olympiad (IPhO):** We evaluate zero-shot transfer on a curated set of mechanics problems from the International Physics Olympiad. We collect and filter problems from 1967–2025 to form an evaluation set of 77 questions. For problems with diagrams, we provide figure captions generated from the original problem context using GPT-4o.

- **HCV (Concepts of Physics):** We evaluate on a set of 512 mechanics problems curated from H. C. Verma's *Concepts of Physics* (Vol.1). For problems with diagrams, we provide figure captions generated from the original problem context using GPT-4o.(Verma, 1999)

- **JEEBench:** A collection of 515 problems from JEE–Advanced (India), covering physics, chemistry, and mathematics, and designed to stress multi-step quantitative reasoning. In our evaluation, we restrict to text-only mechanics questions to avoid confounding gains from visual understanding. We follow the official evaluation pipeline from (Arora et al., 2023)

- **OlympiadBench:** A benchmark of high-difficulty STEM problems sourced from international and national science olympiads. Similar to other real-world evaluations in this section, we focus on text-only mechanics questions when applicable and report exact-match accuracy. We follow the official evaluation pipeline from (He et al., 2024)

- **PHYSICS:** A textbook-derived physics benchmark spanning a range of difficulty levels; only the test set is released publicly. We evaluate on the released test split and restrict to mechanics-related, text-only questions. We follow the official evaluation pipeline from (Zheng et al., 2025b)

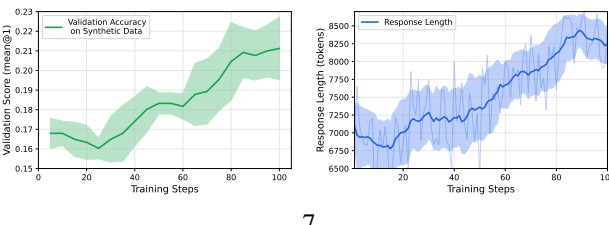

7

*Figure 5.* Validation accuracy (green) versus average response length (blue, in tokens) for Qwen3-30B-Instruct over RL post-training steps. Longer responses are strongly associated with higher validation accuracy, suggesting that post-training encourages more extensive intermediate reasoning.

- **AIME 2025:** We use problems from the 2025 American Invitational Mathematics Examination (AIME) as an out-of-domain math reasoning check. We evaluate using the LightEval (Habib et al., 2023) pipeline and report mean@8 (mean accuracy over 8 sampled responses).(AIME, 2025)

- **MATH 500:** A 500-problem subset of the Hendrycks MATH dataset, which contains competition-style problems with final numeric or symbolic answers. We report exact-match accuracy.(Hendrycks et al., 2021)

**Models:** We evaluate LLMs across multiple model sizes. Specifically, we use Qwen2.5 Instruct checkpoints at 3B, 7B, 14B, and 32B , and additionally include Qwen3-30B-Instruct as a stronger baseline. In our training setup, Qwen3-30B tends to produce substantially longer responses (~8k tokens on average) than comparably sized Qwen2.5 models (~1.5k tokens), which significantly increases RL training cost. Consequently, due to limited compute, we train Qwen3-30B for 100 RL steps, while all Qwen2.5 models are trained for 200 RL steps.

### 3.1. Zero-shot generalization of SIM2REASON

In this section, we evaluate the generalization ability of our SIM2REASON pipeline. We post-train LLMs of different sizes (3B–32B) using RL on our synthetic mechanics questions, and then evaluate the resulting checkpoints on held-out synthetic splits and multiple real-world benchmarks.

Table 1 shows consistent improvements on IPhO Mechanics—up to 7 percentage points across model sizes—despite the fact that the post-training stage uses *no* real-world physics QA data. Notably, the gains persist even for stronger baselines: for example, Qwen3-30B-Instruct improves by +4.4 points on IPhO, suggesting that our synthetic RL signal provides benefits beyond what is already captured by scale and instruction tuning (Figure 5).

Although our default RL training distribution uses numeric questions, we find that the models also improve on other reasoning modes (reverse and symbolic) on the synthetic evaluation splits (Table 1). This indicates that the post-trained models are learning reusable physical reasoning patterns, rather than overfitting to a single question template.

To further test sim-to-real transfer, Table 2 evaluates Qwen2.5-32B on additional real-world physics benchmarks (JEEBench, OlympiadBench, and PHYSICS) as well as out-of-domain math benchmarks (AIME 2025 and MATH 500). We observe consistent gains across all benchmarks. The largest improvement is on JEEBench (+17.9 points), which contains many mechanics questions closely aligned with the phenomena covered by our simulator. We also observe improvements on AIME and MATH, suggesting that training for physics reasoning also strengthens underlying algebraic and multi-step quantitative skills.

In this section, we take a deeper look at the improvements and broader implications of our framework. We first analyze the choice of our post-training training strategy (RL, SFT) and data composition, exploring how our synthetic data compares with existing post-training datasets such as DAPO 17k to improve reasoning. Subsequently, we propose an alternate use case of our framework: using the simulator itself as a scalable benchmarking tool. Finally, we perform a qualitative analysis of the model's outputs to categorize the specific axes of improvement.

### 3.2. Training Strategies for SIM2REASON

SIM2REASON can generate an effectively unbounded number of QA pairs from a physics simulator. A central question then is *how* to distill this simulator-derived supervision into the LLM in a way that (i) improves reasoning, and (ii) preserves the base model's general capabilities, . We investigate two widely used post-training paradigms: (i) supervised fine-tuning (SFT) on high-quality demonstrations, and (ii) reinforcement learning with verifiable rewards (RLVR).

**SFT.** We construct SFT data of 200,000 question-answer pairs by rejection-sampling solutions from strong teacher models (GPT-4, o3, and o4-mini), and then fine-tune the LLM on the resulting trajectories. As shown in Table 3, SFT yields only modest in-distribution gains on our synthetic evaluation and substantially degrades out-of-distribution performance (e.g., -3.9% on IPhO Mechanics). We hypothesize that this is driven by a *large KL shift* from the base Instruct model, which can induce catastrophic forgetting during post-training. This failure mode is consistent with recent analyses showing that aggressive post-training updates can erase general reasoning skills when the optimization signal is narrow or distribution-shifted.(Shenfeld et al., 2025)

**RLVR.** In contrast, RLVR optimizes task success using a sparse, verifiable reward (final-answer correctness), allowing the model to explore diverse solution strategies while

*Table 1.* Performance of Qwen2.5 and Qwen3 family Instruct models before and after RL on synthetic datasets, expressed in percentage. Improvements are shown in parentheses.

| Model | Synthetic Numeric | Synthetic Symbolic | HCV | IPhO Mechanics |
|---|---|---|---|---|
| **Qwen3-30B** | 14.8% | 8.8% | 53.9% | 35.6% |
| + RL (synthetic) | 17.4% (+2.6%) | 8.0% (-0.8%) | 59.0% (+5.1%) | 40.0% (+4.4%) |
| **Qwen2.5-32B** | 8.9% | 5.6% | 50.6% | 19.8% |
| + RL (synthetic) | 21.9% (+13.0%) | 10.4% (+4.8%) | 53.9% (+3.3%) | 25.2% (+5.4%) |
| **Qwen2.5-14B** | 7.0% | 5.6% | 49.3% | 16.07% |
| + RL (synthetic) | 17.0% (+10.0%) | 10.4% (+4.8%) | 51.7% (+2.4%) | 20.45% (+4.4%) |
| **Qwen2.5-7B** | 7.7% | 5.6% | 44.5% | 10.7% |
| + RL (synthetic) | 16.3% (+8.6%) | 9.6% (+4.0%) | 46.3% (+1.8%) | 15.1% (+4.3%) |
| **Qwen2.5-3B** | 4.8% | 3.2% | 31.9% | 5.68% |
| + RL (synthetic) | 12.5% (+7.7%) | 9.4% (+6.2%) | 39.5% (+7.6%) | 13.15% (+7.5%) |

*Table 2.* Mean accuracy of Qwen 2.5 32B Instruct on other real world benchmarks.

| Benchmark | Model | Score |
|---|---|---|
| **JEEBench** | Qwen2.5 32B | 34.38% |
|  | + RL (synthetic) | 52.28% (+17.90%) |
| **PHYSICS** | Qwen2.5 32B | 39.42% |
|  | + RL (synthetic) | 43.09% (+3.67%) |
| **OlympiadBench** | Qwen2.5 32B | 41.41% |
|  | + RL (synthetic) | 44.53% (+3.12%) |
| **AIME 25** | Qwen2.5 32B | 10.83% |
|  | + RL (synthetic) | 12.5% (+1.67%) |
| **MATH 500** | Qwen2.5 32B | 78.4% |
|  | + RL (synthetic) | 82.8% (+4.4%) |

*Table 3.* Comparison of RL vs. SFT on 32B model performance.

| Model (Qwen 32B) | Synthetic | IPhO |
|---|---|---|
| Baseline | 14.0% | 19.8% |
| + SFT | 16.0% (+2.0%) | 15.9% (-3.9%) |
| + RL (Ours) | **32.0%** (+18.0%) | **25.2%** (+5.4%) |

staying closer to the base policy. Empirically, RLVR provides robust improvements both in-distribution (synthetic) and out-of-distribution (real-world benchmarks such as IPhO), suggesting its a more reliable way to distill simulator-derived supervision into generalizable reasoning skills.

### 3.3. Synthetic vs. Real-World Post-Training

The ablations above show that SIM2REASON benefits from careful data design—for example, filtering out shortcut-

solvable questions materially improves transfer. In this section, we ask a broader question: how does simulator-generated post-training compare to *real-world* post-training data? We study this from two complementary angles: first, by comparing against recent open-weight models such as Prime P1 that were post-trained directly on curated real-world physics QA corpora, and second, by comparing against DAPO-17K under a matched RL setup.

*Table 4.* Comparison with open-weight post-trained models trained on real-world QA datasets.

| Base Model | Post-Trained Model | IPhO Accuracy (%) |
|---|---|---|
| Qwen2.5-32B | DAPO-32B | 24.7 |
| Qwen2.5-32B | LIMO-32B | 25.5 |
| Qwen3-30B | Prime P1 30B | 38.6 |
| Qwen3-30B | Sim2Reason (Ours) | **40.**0 |

We first compare against recent open-weight post-trained models that were trained directly on real-world QA corpora (Table 4). Prime P1 (Chen et al., 2025) 30B is trained on over 5,000 curated physics QA pairs from olympiads and textbooks, while DAPO-32B (Yu et al., 2025) and LIMO-32B (Ye et al., 2025) are trained on real-world math QA data. Despite using only simulator-generated synthetic data during post-training, Sim2Reason achieves 40.0% on IPhO, outperforming Prime P1 30B (38.6%), DAPO-32B (24.7%), and LIMO-32B (25.5%). This shows that simulator supervision can compete with, and even exceed, post-training on curated real-world corpora.

Unfortunately, there are currently no publicly available *physics* reasoning post-training datasets that are directly comparable to our setting for a matched data comparison. We therefore next compare against a strong public *math* RL dataset, DAPO-17K, released alongside the DAPO open-source RL system.(Yu et al., 2025) DAPO-17K contains 17K curated mathematical problems designed for outcome-

reward RL training at scale.

Table 5 shows that training on our SIM2REASON synthetic mechanics data yields substantially better IPhO transfer than training on DAPO-17K alone, despite DAPO-17K being an order of magnitude larger than our 1K-sample synthetic subset in this ablation. This suggests that domain-aligned simulator data provides a higher-signal training distribution for physics reasoning than larger but generic math-only corpora.

Combining DAPO-17K with our synthetic data yields a further, albeit smaller, improvement over DAPO-17K alone, indicating partial complementarity: generic math data can strengthen broad quantitative skills, but physics-specific simulator supervision remains the primary driver of IPhO gains.

*Table 5.* Comparison with a real-world post-training dataset.

| Model (Qwen 3B) | IPhO |
| --- | --- |
| Baseline | 5.68 |
| + RL DAPO-17K (Real) | 9.98 |
| + RL Mixed: DAPO-17K (Real) + Synthetic | 10.35 |
| + RL Synthetic (Ours) | **13.15** |

### 3.4. Simulator as a benchmark

Beyond serving as a source of post-training supervision, SIM2REASON also enables a scalable *benchmarking* workflow for scientific reasoning. Measuring progress in physics reasoning is challenging because high-quality real-world evaluation sets are small, expensive to curate, and slow to expand (e.g., olympiad problems require expert selection and careful verification). In contrast, our simulator-driven pipeline can generate large numbers of mechanically grounded questions with automatically verifiable answers, enabling rapid iteration and fine-grained diagnostics across specific phenomena (e.g., pulleys, collisions, springs, rotation).

A key question is whether simulator accuracy predicts real-world reasoning. Figure 9 suggests it does: across models, synthetic accuracy correlates strongly with IPhO mechanics accuracy (Spearman $\rho = 0.79$). This makes simulator-based evaluation a useful proxy for comparing models/ablations and for diagnosing strengths by stratifying results by scene type and physical quantity.

### 3.5. Analysis of Capabilities

In this section, we first analyze the scalability of our pipeline as a data-generation framework, specifically, whether it can be extended to new scene types beyond those currently covered by our DSL. We then analyze the capabilities learned through RL post-training, focusing on three complementary questions: (i) robustness to harder problems, (ii) generalization to questions that cannot be directly simulated in our current environment, and (iii) qualitative changes in solution behavior.

#### 3.5.1. SCALABILITY OF THE PIPELINE

To study scalability, we identify three real-world mechanics questions (from F=ma, USAPhO, and JEE Advanced) that cannot be expressed using our current DSL, and ask an LLM agent to implement them in MuJoCo using two approaches: (1) direct generation of raw MuJoCo XML, and (2) generation within our DSL abstraction space. Direct XML generation succeeds in only 1 out of 3 cases (33%), with failures caused by incorrect spatial reasoning, joint configuration, or missing components. In contrast, DSL-based generation succeeds in all 3 cases (100%), requiring only minor corrections (Figure 30-32). These results suggest that, while direct simulator code is difficult for LLMs to generate reliably, our DSL offers a more effective abstraction for extending the pipeline to new scenarios.

This same abstraction also improves portability. Although our current implementation uses MuJoCo, scene generation operates at the level of entities and connections rather than raw simulator syntax. We therefore ask LLMs to port a subset of DSL entities from MuJoCo to NVIDIA Omniverse (Figure 33), and observe successful transfer for all entities supported by Omniverse's physics engine. Together, these results suggest that broadening the coverage of SIM2REASON does not require rebuilding the entire pipeline from scratch: one can extend the DSL to new scenarios and re-implement entities for new simulator backends while preserving the same overall data-generation workflow.

#### 3.5.2. CAPABILITIES LEARNED THROUGH RL

Having established that the pipeline can scale to new scene types, we next analyze what RL post-training teaches the model, across difficulty levels and broader generalization.

**Coverage Across Difficulty Levels.** We evaluate robustness across difficulty tiers in the PHYSICS benchmark. As shown in Table 6, RL post-training on SIM2REASON improves performance at *every* tier.

Gains are modest at lower tiers (e.g., +2.8% at High School and Below) and largest at the Postgraduate tier (+5.6%), suggesting simulator-based RL particularly strengthens harder multi-step quantitative reasoning. We use Gemini 2.5 Flash as a verifier.

**Generalization Beyond Simulation.** A key question is whether the gains of Sim2Reason are limited to scenarios were explicitly model in MuJoCo. We find that improvements transfer to problems that are *not* directly covered by our current library of entities. In principle, many such

*Table 6.* Detailed performance across difficulty levels on the PHYSICS benchmark.

| Category | Qwen 32B | + RL (synthetic) |
|---|---|---|
| High School and Below | 65.5% | 68.3% (+2.8%) |
| High School Olympiad | 52.9% | 54.0% (+1.1%) |
| Undergraduate | 47.9% | 48.4% (+0.5%) |
| Postgraduate | 32.2% | 37.8% (+5.6%) |

### IPhO 2012 Q1: Focus On Sketches

A ball is thrown with a fixed initial speed $v_0$ in a homogeneous gravitational field. The x-axis is horizontal, and the z-axis is vertical (opposing gravity $g$).

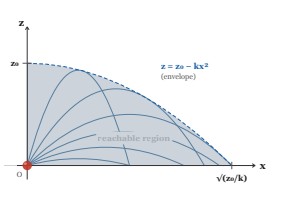

By adjusting the launching angle, the ball can hit any target within a region defined by the parabola: $z \leqslant z_0 - kx^2$

You can use this fact without proving it. Find the constants $z_0$ and $k$ in terms of $v_0$ and $g$.

---

**△ BASE MODEL**

**Step 1:** Write Equations of Motion
The trajectory is given by:
$x = v_0 \cos(\theta) t$
$z = v_0 \sin(\theta) t - \frac{1}{2} g t^2$

Eliminating $t$, we get the trajectory equation:
$z = x \tan(\theta) - \frac{g x^2}{2 v_0^2 \cos^2(\theta)}$

**Step 2:** Match Coefficients
We compare this specific trajectory to the given boundary equation:
$z = z_0 - kx^2$.

To find the constants, we assume the standard case for maximums, which is $\theta = 45°$.
- At 45 degrees, $\tan(\theta) = 1$.
- At 45 degrees, $\cos^2(\theta) = 0.5$.

**Step 3:** Solve for $k$
Looking at the $x^2$ term in the trajectory equation:
Coefficient $= \frac{g}{2 v_0^2 \cos^2(\theta)}$
Substitute $\cos^2(45) = 0.5$:
$k = \frac{g}{2 v_0^2 \cdot 0.5}$
$k = g/v_0^2$

> **ERROR:** Assumed boundary is defined by the 45-degree path.

**Conclusion:**
$z_0 = v_0^2/(2g)$
$k = g/v_0^2$

**Predicted Answer:**
$z_0 = v_0^2/2g, k = g/v_0^2$

---

**✓ RL-FINETUNED MODEL**

**Step 1:** Analyze the Physical Meaning
The boundary conditions are:
1. The maximum possible height ($z_{max}$) where $x = 0$.
2. The maximum possible range ($x_{max}$) where $z = 0$.

**Step 2:** Find $z_0$ (The Peak)
The maximum height is achieved by throwing the ball straight up ($\theta = 90°$).
Using energy conservation ($mgh = \frac{1}{2}mv^2$):
$z_0 = v_0^2/(2g)$

**Step 3:** Find $k$ (The Width)
We know max range occurs at 45°:
$x_{max} = v_0^2/g$
Now, substitute this point ($z = 0, x = v_0^2/g$) into the boundary equation:
$0 = z_0 - k x_{max}^2$
$0 = \frac{v_0^2}{2g} - k (\frac{v_0^2}{g})^2$

> **CORRECT:** Assumed the correct boundary condition.

**Step 4:** Solve for $k$
$k \frac{v_0^4}{g^2} = \frac{v_0^2}{2g}$
$k = \frac{v_0^2}{2g} \cdot \frac{g^2}{v_0^4} = \frac{g}{2v_0^2}$

**Conclusion:**
$z_0 = v_0^2/(2g)$
$k = g/(2v_0^2)$

**Predicted Answer:**
$z_0 = v_0^2/2g, k = g/2v_0^2$

*Figure 6.* **LLM answers before (left) and after (right) RLVR.**
Question adapted from IPhO 2012 Question 1 "Focus on sketches".

problems could be simulated, but doing so can require *bespoke* entity design and scene construction tailored to that specific setting (e.g., adding specialized celestial-body interactions).

For example, the problem in Figure 25 involves a rocket taking off a planet in the presence of a star. Accurately simulating this setup would require implementing additional entity logic with this exact case in mind. Nonetheless, the base Qwen2.5-32B-Instruct model fails to solve the problem in any of eight trials, whereas after RL on our synthetic data the success rate increases to 50% (4/8). This suggests that the post-trained model is learning transferable abstractions (e.g., formulating constraints and bookkeeping forces/energy), rather than merely overfitting to simulated scenes.

**Qualitative Examples.** To concretely illustrate these gains, we present comparative case studies across real-world problems. We observe improvements along several axes: **arithmetic** (reducing calculation errors; Figures 28, 29), **physical reasoning** (mapping text to correct equations and boundary conditions; Figures 6, 25, 27), and **strategic planning** (e.g., unit conversions and intermediate checks; Figure 26).

## 4. Conclusion

We presented SIM2REASON, a simulator-driven pipeline that procedurally generates diverse physics scenes, converts simulated traces into verifiable QA pairs, and post-trains LLMs with RLVR. Across multiple real-world benchmarks (e.g., IPhO mechanics), models trained only on synthetic simulator supervision show consistent zero-shot sim-to-real gains, suggesting simulators are a scalable source of reasoning supervision.

A direct avenue for future work is to combine simulator-generated data with curated real-world QA to further improve robustness and coverage. More broadly, extending this approach beyond classical mechanics to other areas of physics (e.g., E&M, thermodynamics) and to other physical sciences is a promising direction.

## Impact Statement

This work investigates training language models for physical reasoning using synthetic question–answer supervision generated from physics simulators. We expect the primary positive impact to be improved access to high-quality scientific tutoring and problem-solving tools, and a reduction in dependence on scraping internet QA data.

Potential risks include misuse of stronger reasoning models (e.g., to assist in harmful engineering) and over-reliance on simulator-generated supervision, which may encode mod-

eling assumptions and failure modes that do not hold in the real world. To mitigate these issues, we emphasize evaluation on real-world benchmarks, report limitations of simulator fidelity and coverage, and encourage downstream deployments to include safeguards, monitoring, and domain-specific validation.

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

# A. Related Work

**Reinforcement Learning from Verifiable Feedback**    Recent work has explored Reinforcement Learning from Verifiable Rewards (RLVR) as a scalable alternative to human preference annotation for training reasoning-capable language models (DeepSeek-AI et al., 2025; Yu et al., 2025; Shao et al., 2024; Yang et al., 2025). In RLVR, models are trained using automatically verifiable signals—such as exact-answer matching, program execution, theorem proving, or symbolic checks—to provide dense, objective reward signals for complex reasoning tasks (Zhu et al., 2024; Xin et al., 2024; Yang et al., 2025). This paradigm has been successfully applied in domains such as mathematics, code generation, and formal reasoning, where correctness can be algorithmically verified. However, existing RLVR approaches rely on domains with deterministic and symbolic verification pipelines and are limited by the availability of structured ground truth problems and answers. In contrast, our work extends the RLVR paradigm to physical reasoning, where supervision is derived from physics simulation rather than question-answer pairs. By using simulators to generate verifiable outcomes and synthetic QA supervision, we enable RL-based training of LLMs in domains where formal verification might be infeasible, demonstrating zero-shot transfer to real-world physics benchmarks such as IPhO.

**Symbolic Regression**    Symbolic regression aims to recover interpretable physical laws from data (Angelis et al., 2023), using methods ranging from genetic programming (Schmidt and Lipson, 2009) to sparse regression (Brunton et al., 2016) and neural approaches (Udrescu and Tegmark, 2020; Raissi et al., 2019). While these methods are appealing for their interpretability, purely data-driven symbolic regression is often most successful in relatively simple, low-dimensional settings and can become brittle when the data are noisy, incomplete, or high-dimensional without additional inductive bias or physical constraints (Angelis et al., 2023; Reinbold et al., 2021). Recent work suggests that LLMs may help address some of these limitations by proposing candidate functional forms, programs, or symbolic expressions that guide the search over equations more effectively than brute-force enumeration alone (Shojaee et al., 2025).

**Synthetic Data training**    Synthetic data has emerged as a powerful alternative to manual annotation across several domains. In mathematics, AlphaGeometry procedurally generates large-scale synthetic geometry training data to solve Olympiad-level geometry problems (Trinh et al., 2024). In robotics, *Solving Rubik's Cube with a Robot Hand* shows that large-scale simulator training with automatic domain randomization can transfer complex manipulation policies to the real world (Akkaya et al., 2019). In language-model post-training, methods such as MetaMath and Self-Instruct use LLMs to synthetically expand instruction or reasoning datasets (Yu et al., 2023; Wang et al., 2023), while more recent approaches such as PAPRIKA and SPIRAL use synthetic interaction data or self-play to create scalable training curricula without relying exclusively on human-written supervision (Tajwar et al., 2025; Liu et al., 2025).

Our work is complementary to these efforts but targets a distinct setting. Our setting is different from prior synthetic-data work in math and code, where the underlying tasks already come with clean symbolic structure and canonical forms of supervision. In physics, by contrast, the goal is often to answer natural-language questions about systems governed by continuous dynamics. We therefore propose a recipe for turning physics simulators into generators of structured post-training data—including English QA pairs with automatically verifiable answers—and show that this synthetic supervision alone yields zero-shot gains on real-world physics benchmarks.

# B. Domain-Specific Language and Timestep pruning strategy

We summarize the two additional components used to build training data. Figure 7 shows the YAML-based scene-generation DSL and an example MuJoCo rendering produced by compiling it to MuJoCo XML, while Figure 8 illustrates our timestep-pruning heuristic that removes unstable trace suffixes before QA generation.

# C. Additional Results

### C.1. Correlation between simulator eval and real

Figure 9 reports a correlation analysis across models/runs, showing that higher accuracy on our SIM2REASON synthetic questions tends to coincide with higher accuracy on IPhO mechanics. This supports using the synthetic QA suite as a lightweight proxy for real-world physics reasoning performance.

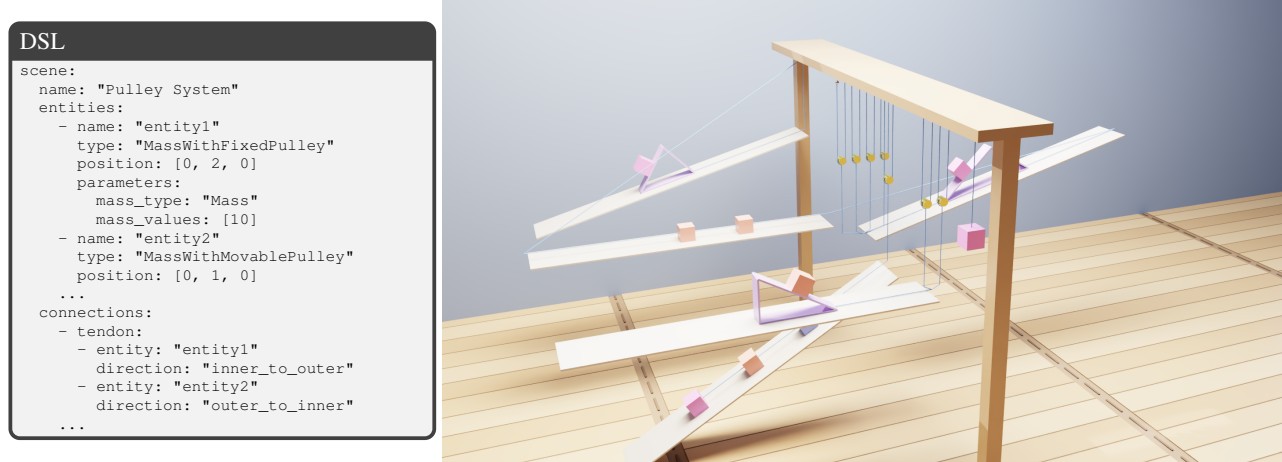

*Figure 7.* Example of our scene-generation DSL (top) and the corresponding simulator-rendered scene produced by compiling the DSL into simulator scene (bottom). The DSL composes scenes from reusable entities and bodies with explicit connection modes, enabling scalable procedural generation while restricting randomization to physically meaningful parameters.

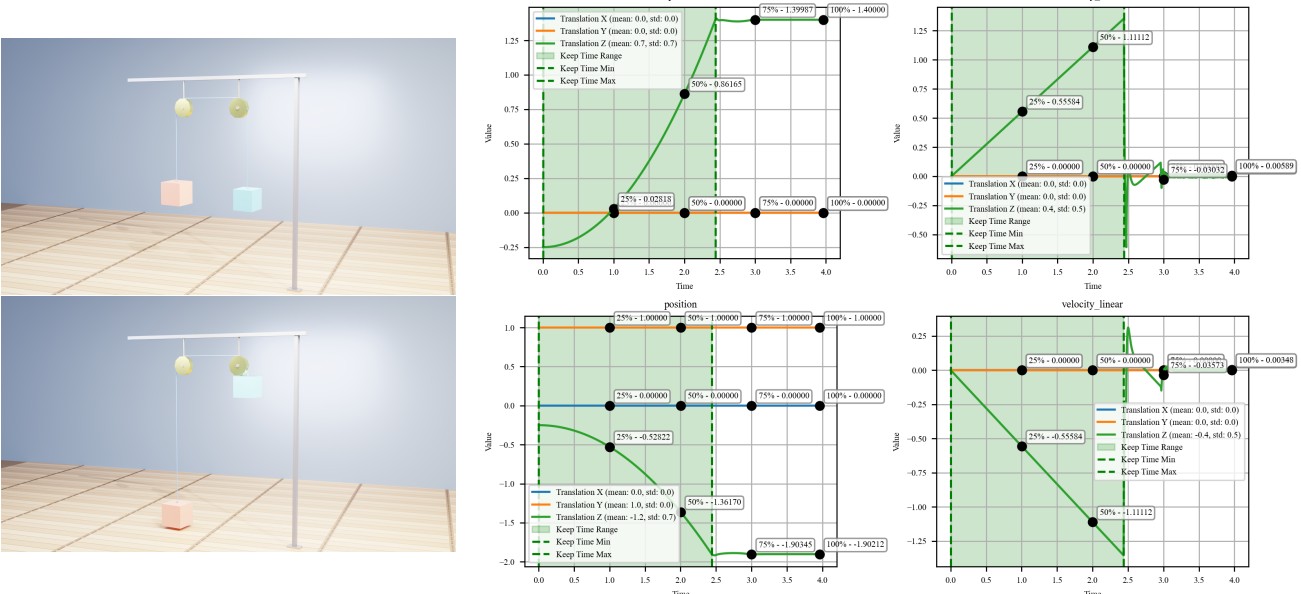

*Figure 8.* Timestep pruning for simulation traces with unmodelled transitions. Left: MuJoCo scene snapshots at the start and at time 3s. Right: recorded time-series signals; when a sliding-window deviation criterion flags an outlier (e.g., due to contact between block and pulley), we keep only the stable prefix (green) and discard the remainder before generating QA pairs.

### C.2. Ablations: QA format and data filtration

We ablate two design choices in our synthetic RL pipeline: (i) the *question format* used during post-training (Section 2.3), and (ii) whether we apply the *shortcut-solution* filtering described in Section 2.4. Unless stated otherwise, we report IPhO Mechanics accuracy for Qwen2.5-3B Instruct.

**QA format:** We compare training with numeric questions (our default) against reverse and symbolic variants. Table 7a shows that numeric QA yields the strongest transfer to IPhO.

**Shortcut filtering:** We also test the impact of removing shortcut-solvable questions via scene ablations. As shown in Table 7b, shortcut filtering is critical: training without filtering yields substantially smaller gains than training on the filtered numeric distribution.

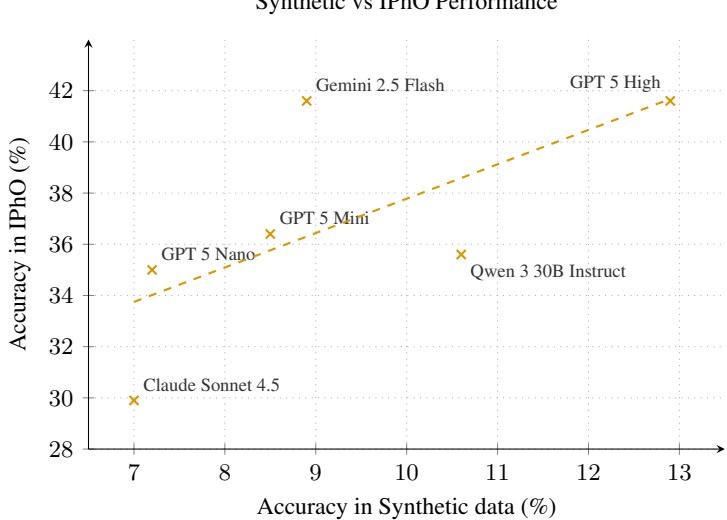

*Figure 9.* Correlation between accuracy on SIM2REASON synthetic questions and IPhO mechanics questions.

*(a)* Improvements by each QA format during RL post-training.

| Model (Qwen 3B) | IPhO |
|---|---|
| Baseline | 5.68% |
| + RL (reverse) | 5.84% |
| + RL (symbolic) | 7.46% |
| + RL (numeric) | **13.15%** |

*(b)* Effect of shortcut-solution filtering during data-generation.

| Model (Qwen 3B) | IPhO |
|---|---|
| Baseline | 5.68% |
| + RL (*no* filter) | 7.14% |
| + RL ( filtered) | **13.15%** |

*Table 7.* Ablations on (a) QA format and (b) Data filtration

## D. Bodies and their parameters

We define a list of bodies, along with their randomizable parameters in Table 8.

## E. Recorded physical quantities

During simulation, we log time-series data for each scene to enable question-answer pair generation. We group data into three categories: **mass**-related (body state and dynamics), **string**-related (length/tension), and **contact** (interaction forces). We list the recorded quantities for each category in Table 9.

## F. Entities and their Connections

Here, we show a list of entities that we define (Figures 10–24). The randomizable parameters for each entity are visualized in the figures by their respective mathematical notations. The connection points and modes are also visualized as dotted lines.

| Body | Symbol(s) | Description |
|---|---|---|
| Mass | $m$ | Point mass / block mass. |
| Sphere | $r$, $m$ | Sphere radius and mass. |
| Polygonal prism | $n$, $r$, $h$, $m$ | Number of sides, circumscribed radius, height, and mass. |
| Cylinder | $r$, $h$, $m$ | Cylinder radius, height, and mass. |
| Disc | $r$, $m$ | Disc radius and mass. |
| Bar | $w$, $\ell$, $h$, $m$ | Bar width, length, height, and mass. |
| Hemisphere | $r$, $m$ | Hemisphere radius and mass. |
| Bowl | $r$, $h_c$, $t$, $m$ | Bowl radius, cutting-plane height $h_c$, shell thickness $t$ (if hollow), and mass. |
| Sphere with spherical hole | $r$, $r_h$, $p_h$, $t$, $m$ | Outer radius $r$, hole radius $r_h$, hole position $p_h$, shell thickness $t$ (if hollow), and mass. |
| Rocket | $m_{\mathrm{dry}}$, $m_0$ | Dry mass $m_{\mathrm{dry}}$ and initial total mass $m_0$. |
| Triangular prism | $\alpha_L$, $\alpha_R$, $m$ | Left/right face slopes (angles) and mass. |
| Plane | $\alpha$ | Plane slope (incline angle). |
| Pulley | $m$ | Pulley mass. |
| Spring–mass system | $\{k_i\}$, $\{\ell_{0,i}\}$, $\{x_i\}$, $\{m_i\}$ | Spring constants, natural lengths, mass positions, and masses connected by springs. |

*Table 8.* Bodies used by the DSL and the corresponding randomizable parameters.

**mass_with_fixed_pulley** consists of a fixed pulley with one side open for connection to other entities (represented by dotted line), and the other connected to a simple mass system. Below are the 3 variants of mass systems which are supported by this entity.

**ENTITY VISUALIZATION**

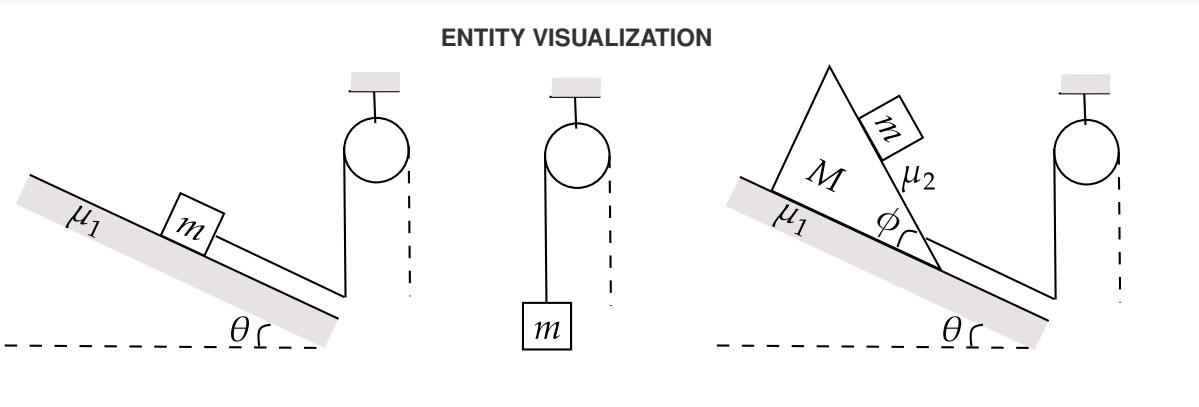

*Figure 10.* Mass With Fixed Pulley

| Category | Quantity | Description |
|---|---|---|
| Mass | displacement | Body displacement / position (in world frame). |
| Mass | com_offset | Vector from body frame origin to center of mass. |
| Mass | velocity (6D) | Linear and angular velocity. |
| Mass | acceleration (6D) | Linear and angular acceleration. |
| Mass | mass | Body mass. |
| Mass | momentum (6D) | Linear and angular momentum. |
| Mass | net force (6D) | Net force/torque (consistent with $F = ma$). |
| Mass | kinetic_energy_linear | Translational kinetic energy. |
| Mass | kinetic_energy_angular | Rotational kinetic energy. |
| Mass | potential_energy | Gravitational potential energy. |
| Mass | inertia | Inertia tensor. |
| Mass | em_potential_energy | Electromagnetic potential energy (when applicable). |
| Contact | normal_force | Normal contact force at interaction points. |
| Contact | friction_force | Tangential/frictional contact force. |
| String | length | Current string length. |
| String | velocity | Rate of change of string length. |
| String | force | Tension force. |
| String | stiffness | Spring constant (for elastic strings/springs). |

*Table 9.* Physical quantities recorded from MuJoCo for each simulated scene.

**mass_with_movable_pulley** consists of a movable pulley with both sides connected to one of the variants of **mass_with_fixed_pulley** (represented by dotted shapes $E_1$ and $E_2$), and the top is open for connection to other entities (represented by dotted line).

**ENTITY VISUALIZATION**

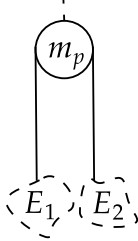

*Figure 11.* Mass With Movable Pulley

**mass_with_reverse_movable_pulley** is the reverse variant of **mass_with_movable_pulley** where the two connections of the pulley pull it up, whereas in **mass_with_movable_pulley** the two connections of the pulley pull it down.

**ENTITY VISUALIZATION**

*Figure 12.* Mass With Movable Pulley

**two_side_mass_plane** consists of a mass on plane which can be connected to other entities on either sides.

**ENTITY VISUALIZATION**

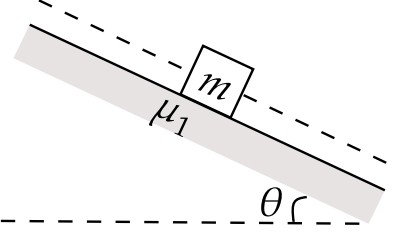

*Figure 13.* Two Side Mass Plane

**stacked_mass_plane** consists of long mass blocks stacked on top of each other on a plane. Each of these mass blocks can be connected to other entities on either side.

**ENTITY VISUALIZATION**

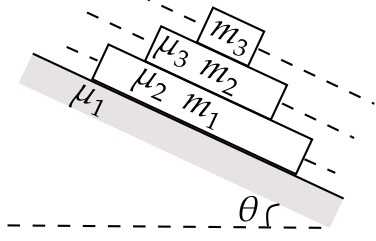

*Figure 14.* Stacked Mass Plane

`directed_mass` consists of mass block suspended from two fixed pulleys. The other ends of each of these pulleys can be connected to other entities.

**ENTITY VISUALIZATION**

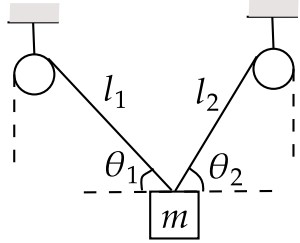

*Figure 15.* Directed mass

`mass_prism_plane` consists of a movable inclined plane and two mass blocks on either side of it. These mass blocks are connected to each other by a string.

**ENTITY VISUALIZATION**

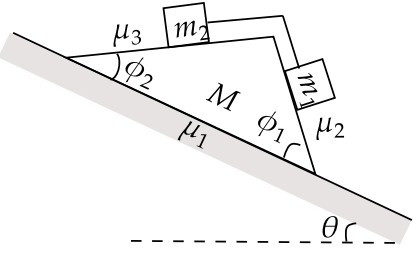

*Figure 16.* Mass Prism Plane

`mass_box_plane` consists of a large movable mass block and optional mass blocks on either face of it. These mass blocks are connected to each other by a string.

**ENTITY VISUALIZATION**

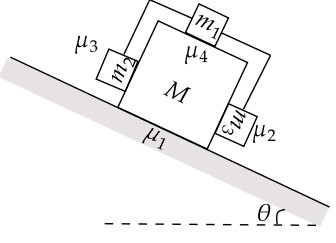

*Figure 17.* Mass Box Plane

**twoD_collision_plane** consists of a large frictionless plane and a couple of spheres on top it, each given with some initial velocity.

**ENTITY VISUALIZATION**

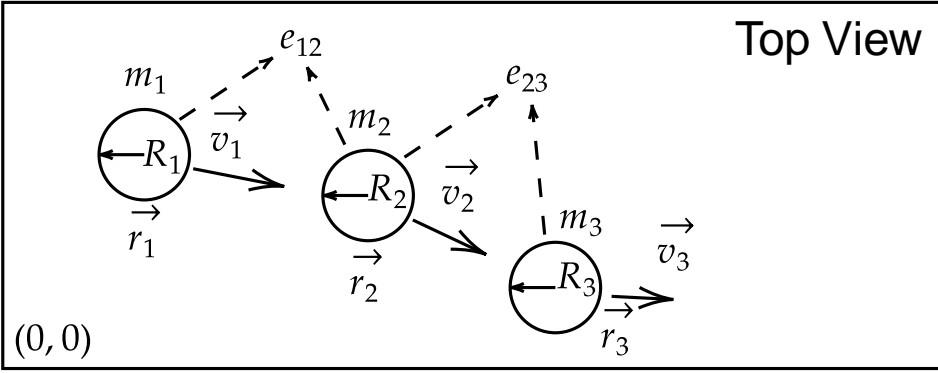

*Figure 18.* TwoD Collision Plane

**complex_collision_plane** consists of a long frictionless plane and a couple of objects on top it, each given with some initial velocity. This setup is entirely 1D to lower complexity of the problems. Possible objects are sphere, block, fixed wall and spring blocks.

**ENTITY VISUALIZATION**

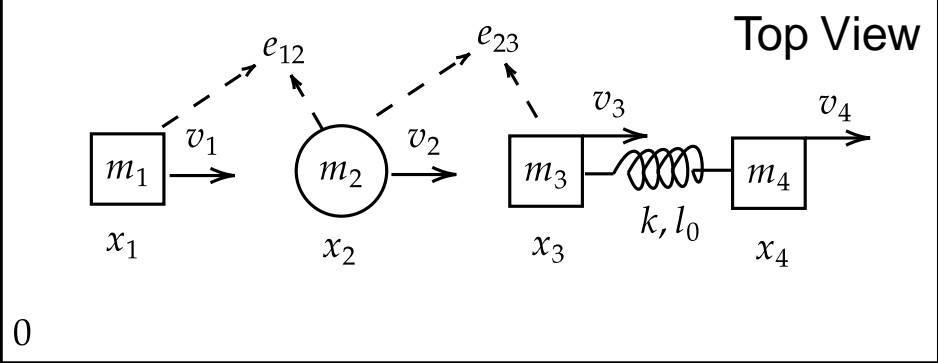

*Figure 19.* Complex Collision Plane

`solar_system` consists of a stationary star and a couple of planets revolving around it.

**ENTITY VISUALIZATION**

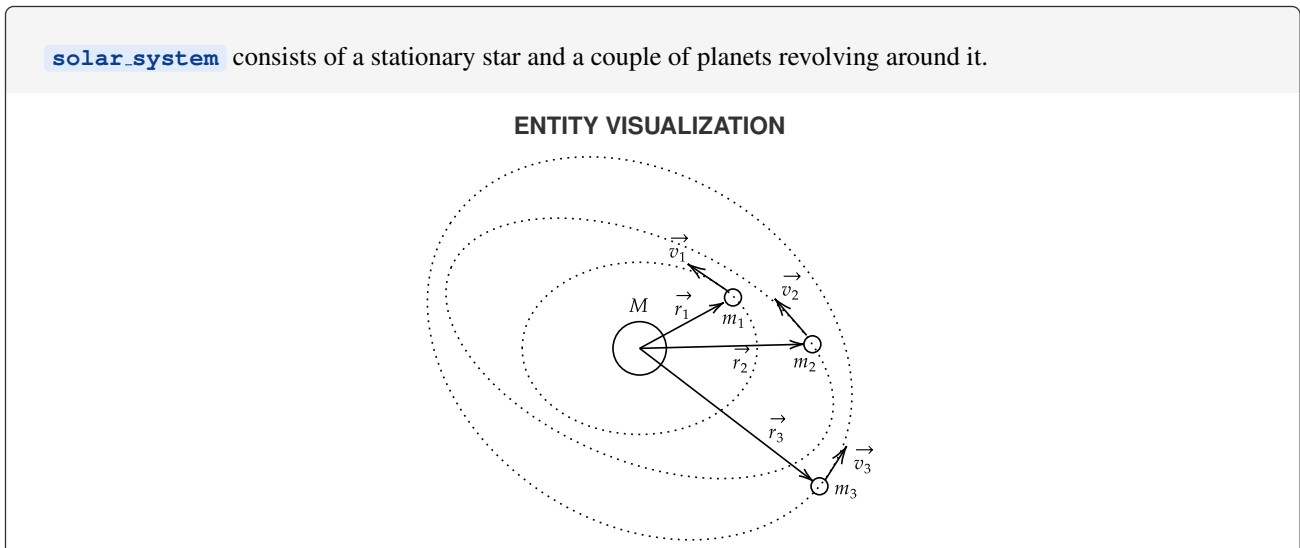

*Figure 20.* Solar System

`rocket_entity` consists of a stationary planet and a rocket taking off of the planet. The rocket has a dry mass $m_0$ and initial mass $m$. It burns fuel to propel itself, losing mass at a rate of $\mu$.

**ENTITY VISUALIZATION**

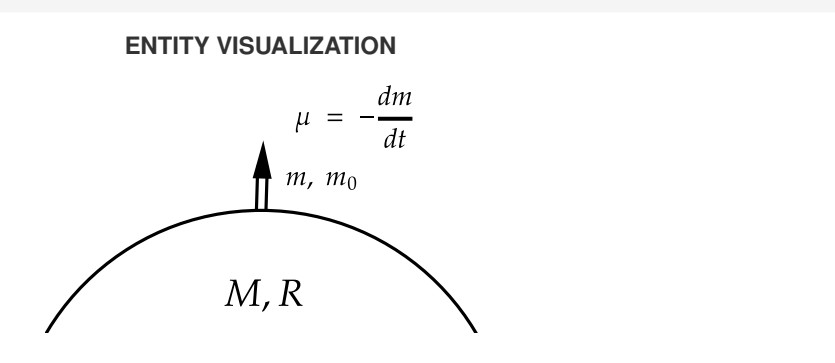

*Figure 21.* Rocket Entity

`rotation_entity` consists of multiple 3D shapes attached to each other with rigid joints so that they move together. Additionally, they are attached to a pivot, allowing them to rotate around it due to gravity in a pendulum motion.

**ENTITY VISUALIZATION**

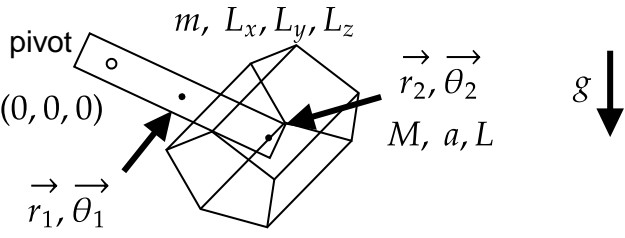

*Figure 22.* Rotation Entity

`rolling_entity` consists of 3D shapes rolling on an inclined plane. We choose primitive 3D shapes such as spheres, cylinder and polygon cylinders. Additionally we also randomly cutout a shape from the body-for instance cutting a smaller sphere from a sphere results in a spherical shell. We automate this by using blender to generate arbitrary cutout shapes.

**ENTITY VISUALIZATION**

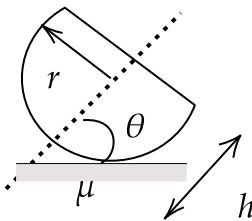

*Figure 23.* Rolling Entity

`em_entity` consists of a moving charged particle in space in the presence of varying Electric ($\vec{E}$) and Magnetic fields ($\vec{B}$). These fields vary as a known function of the position, velocity of the particle and time.

**ENTITY VISUALIZATION**

*Figure 24.* Electro Magnetism Entity

## JEE ADVANCED 2017 PAPER 2

A rocket is launched **normal to the surface of the Earth**, away from the Sun, along the line joining the Sun and the Earth.

**Parameters:**
- Mass of Sun = $300,000\times$ Mass of Earth
- Distance = $25,000\times$ Radius of Earth
- Escape Velocity (Earth) = $11.2$ km/s

The minimum initial velocity ($v_S$) in km/s required for the rocket to be able to leave the Sun-Earth system is closest to:

[A] $22$    [B] $42$    [C] $62$    [D] $72$

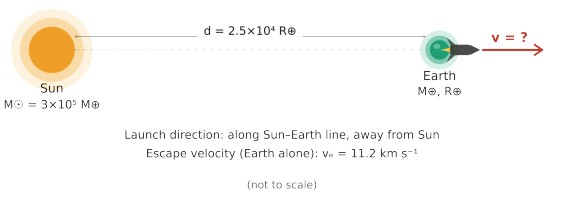

$d = 2.5\times10^4\ R\oplus$

Sun
$M\odot = 3\times10^5\ M\oplus$

Earth
$M\oplus,\ R\oplus$

v = ?

Launch direction: along Sun-Earth line, away from Sun
Escape velocity (Earth alone): $v_e = 11.2$ km s$^{-1}$

(not to scale)

### △ BASE MODEL

**Step 1:** Calculate Escape Velocity from Sun
Using the given ratios ($M_{\text{Sun}} = 3 \times 10^5 M_E$, $d = 2.5 \times 10^4 R_E$):

$$v_{\text{sun}} = \sqrt{\frac{2GM_{\text{Sun}}}{d}} \approx 54.78 \text{ km/s}$$

**ERROR: Correct ≈ 39 km/s**

**Step 2:** Calculate Total Velocity
To escape both bodies, we sum velocities:

$$v_{\text{total}} = 11.2 + 54.78 \approx 65.98 \text{ km/s}$$

**ERROR: Linear addition**
Treats gravitational potentials as independent additive quantities (physically wrong).

**Conclusion:**
The calculated value ($66$ km/s) is closest to [C].

**Predicted Answer: [C] 62 km/s**

### ✓ RL-FINETUNED MODEL

**Step 1:** Analyze the System
We need to escape the combined potential of Earth and Sun.
- Earth Escape: $11.2$ km/s
- Sun Escape (at Earth's distance): $\approx 42$ km/s

**Step 2:** Determine Effective Velocity
Since the Sun's gravitational well is significantly deeper, the escape velocity is dominated by the Sun.

$$v_{\text{total}} = \sqrt{v_{\text{sun}}^2 + v_{\text{earth}}^2}$$

**Correct: Energy Addition**

Because $v_{\text{sun}}(42) \gg v_{\text{earth}}(11)$, the result is approximately $v_{\text{sun}}$.
$v_{\text{total}} \approx 42$ km/s
**Conclusion:**
The minimum velocity is dictated by the Sun.

**Predicted Answer: [B] 42 km/s**

*Figure 25.* **LLM answers before (left) and after (right) RL finetuning.** Question adapted from JEE Advanced 2017 Paper 2.

---

### IPhO 2005 Q1: An Ill Fated Satellite

A geosynchronous satellite of mass $m$ is in a circular orbit of radius $r_0$ with velocity $v_0$.

An error causes the engine to fire, providing an instantaneous radial thrust $\Delta v$ directed towards Earth. We characterize this boost by the parameter:

$$\beta = \frac{\Delta v}{v_0}$$

**Task:** Calculate the minimum boost parameter $\beta_{esc}$ needed for the satellite to escape Earth's gravity.

---

#### △ BASE MODEL

**Step 1:** Orbital Velocity

$v_0 = \sqrt{GM_E/r_0}$

**Step 2:** Escape Condition

Escape velocity: $v_{esc} = \sqrt{2GM_E/r_0}$
Condition: $v_{after} \geqslant v_{esc}$

Since the thrust is directed radially inward, the effective velocity becomes reduced:
$v_{after} = v_0 - \Delta v$

> **ERROR: Linear subtraction.**
> Fails to treat velocity as a vector (radial $\perp$ tangential).

**Step 3:** Solve for $\beta$

$\sqrt{GM/r} - \beta\sqrt{GM/r} \geqslant \sqrt{2GM/r}$

$(1 - \beta) \geqslant \sqrt{2} \implies \beta \approx -0.41$

(Model confuses signs and algebra)

$\beta \geqslant \sqrt{2} - 1 \approx 0.41$

**Final Answer:**

$\boxed{\beta_{esc} = \sqrt{2} - 1 \approx 0.41}$

#### ✓ RL-FINETUNED MODEL

**Step 1:** Energy Analysis

Initial Energy: $E_0 = -GMm/2r_0$
Escape Velocity needed: $v_{esc} = \sqrt{2GM/r_0}$

**Step 2:** Vector Addition

The kick is radial ($\Delta v$), perpendicular to the orbital velocity ($v_0$). The new velocity magnitude adds in quadrature:

$v_{new} = \sqrt{v_0^2 + (\Delta v)^2}$

> **Correct: Quadrature Addition**

**Step 3:** Escape Condition

$v_{new} \geqslant v_{esc}$
$\sqrt{v_0^2 + (\Delta v)^2} \geqslant \sqrt{2}v_0$

Substitute $\Delta v = \beta v_0$:
$\sqrt{v_0^2 + \beta^2 v_0^2} \geqslant \sqrt{2}v_0 \implies \sqrt{1 + \beta^2} \geqslant \sqrt{2}$

**Step 4:** Solve for $\beta$

$1 + \beta^2 \geqslant 2 \implies \beta^2 \geqslant 1 \implies \beta \geqslant 1$

**Final Answer:**

$\boxed{\beta_{esc} = 1}$

*Figure 26.* **LLM answers before (left) and after (right) RL finetuning.** Question adapted from IPhO 2005 Q1 "An Ill Fated Satellite".

**JEE Advanced 2023 Paper 1**

A small mass ($m = 0.10$ kg) moving with initial speed $u = 5.00$ m/s collides elastically with a bar (Mass $M = 1.00$ kg, Length $L = 0.20$ m) that is pivoted at one end. The collision occurs at a distance $L/2$ from the pivot.

**Parameters:**
- $m = 0.10$ kg; $u = 5.00$ m/s
- $M = 1.00$ kg; $L = 0.20$ m
- $I_{bar}$ (about pivot) $= ML^2/3$

The velocity of the small mass ($v$) after the collision is closest to:
  **[A]** $4.30$ m/s   **[B]** $5.00$ m/s   **[C]** $-2.50$ m/s   **[D]** $2.15$ m/s

---

△ **BASE MODEL**

**Step 1:** Conservation of Angular Momentum
$L_{initial} = m \cdot u \cdot (L/2)$
$L_{initial} = 0.1 \times 5.0 \times 0.1 = 0.005$

**ERROR**: Arithmetic (Should be 0.05)

$L_{final} = I_{pivot}\omega + mv(L/2)$
$0.005 = 0.00133\omega + 0.005v$

**Step 2:** Conservation of Kinetic Energy
Since collision is elastic, conserve energy.
$KE_{initial} = 0.5mu^2 = 1.25$ J

For the bar, we consider both rotation and translation:
$KE_{bar} = 0.5I_{pivot}\omega^2 + 0.5Mv_{cm}^2$

**ERROR**: Double counting energy
  (Using $I_{pivot}$ includes energy of COM. Adding linear KE counts twice.)

**Step 3:** Solve System
The model attempts to fit the values... finds $v \approx 4.30$ m/s, but associates it with wrong option due to internal confusion.
**Predicted Answer: [B] 5.00 m/s**

---

✓ **RL-FINETUNED MODEL**

**Step 1:** Conservation of Angular Momentum
Calculate initial angular momentum correctly:
$L_{initial} = m \cdot u \cdot (L/2)$
$L_{initial} = 0.10 \times 5.00 \times 0.10 = 0.05$

**Correct: Arithmetic**

$L_{final} = I_{pivot}\omega + mv(L/2)$
$0.05 = 0.0133\omega + 0.01v$

**Step 2:** Conservation of Kinetic Energy
For a bar pivoting at the end, KE is purely rotational about pivot:
$KE_{bar} = 0.5I_{pivot}\omega^2$

**Correct: Energy counted correctly**

Equation:
$1.25 = 0.05v^2 + 0.00665\omega^2$

**Step 3:** Solve System
Using Momentum equation and substituting into Energy...
Solving yields $\omega \approx 5.0$ rad/s and $v = 5 - 1.33(5.0) \approx 4.30$ m/s.
**Predicted Answer: [A] 4.30 m/s**

*Figure 27.* **LLM answers before (left) and after (right) RL finetuning.** Question adapted from JEE Advanced 2023 Paper 1.

**IPhO 2018 Q1: LIGO-GW150914**

Calculate the dimensionless coefficient $\xi$ for the power $\mathcal{P}$ emitted in gravitational waves by a binary system.

**Formula:**

$$\mathcal{P} = \frac{G}{5c^5} \sum_{i,j} \left( \frac{\mathrm{d}^3 Q_{ij}}{\mathrm{d}t^3} \right)^2 = \xi \frac{G}{c^5} \mu^2 L^4 \Omega^6$$

**Given Quadrupole Moments** (where $k = 2\Omega$):
$Q_{ii} = \frac{\mu L^2}{2}(a_i + b_i \cos kt)$  and  $Q_{ij} = \frac{\mu L^2}{2} c_{ij} \sin kt$  (for $i \neq j$)

**Parameters:**
$b_1 = 1, b_2 = -1, b_3 = 0$;  $c_{12} = c_{21} = 1$ (others 0).

The value of $\xi$ is:  **[A]** 1.6  **[B]** 3.2  **[C]** 6.4  **[D]** 12.8

---

### △ BASE MODEL

**Step 1:** Compute Third Time Derivatives
We differentiate the quadrupole moments three times.
$Q(t) \sim \frac{1}{2} \cos(2\Omega t)$

First deriv: $\sim \frac{1}{2} \cdot (2\Omega) = \Omega$ Second deriv:
$\sim \Omega \cdot (\Omega) = \Omega^2$
Third deriv: $\sim \Omega^2 \cdot (2\Omega) = 2\Omega^3$

| **ERROR**: Failed chain rule (missing factor)

Resulting Coefficient: $A = 2$

**Step 2:** Square and Sum
We square the coefficient found in Step 1.
Square: $A^2 = 2^2 = 4$

Summing over the non-zero components:
Sum = 4 (from diagonal) +4 (from off-diagonal)
= 8

**Step 3:** Apply Formula
$\mathcal{P} = \frac{1}{5} \cdot$ Sum $\cdot$ (constants)
$\xi = \frac{8}{5} = 1.6$
**Predicted Answer: [A] 1.6 (Incorrect)**

---

### ✓ RL-FINETUNED MODEL

**Step 1:** Compute Third Time Derivatives
We apply the chain rule consistently for the third derivative.
$Q(t) \sim \frac{1}{2} \cos(2\Omega t)$

First deriv: $\frac{1}{2} \cdot (-2\Omega) \sin(2\Omega t)$
Second deriv: $\frac{1}{2} \cdot (-4\Omega^2) \cos(2\Omega t)$
Third deriv: $\frac{1}{2} \cdot (8\Omega^3) \sin(2\Omega t)$

| **Correct: Consistent Chain Rule**

Resulting Coefficient: $A = 4$

**Step 2:** Square and Sum
We square the coefficient found in Step 1.
Square: $A^2 = 4^2 = 16$

Summing over the components (using
$\sin^2 + \cos^2 = 1$ identity):
Sum = 16 (from diagonal) +16 (from off-diagonal)
= 32

**Step 3:** Apply Formula
$\mathcal{P} = \frac{1}{5} \cdot$ Sum $\cdot$ (constants) $= \frac{32}{5} = 6.4$
**Predicted Answer: [C] 6.4 (Correct)**

---

*Figure 28.* **LLM answers before (left) and after (right) RL finetuning.** Question adapted from IPhO 2018 Question 1 "LIGO-GW150914".

---

### IPhO 2013 Q1

A meteoroid from Comet Encke impacted Earth (the Maribo meteorite). Measurements show the meteorite formed shortly after the solar system's birth.

In its orbit around the Sun, the minimum and maximum distances between Comet Encke and the Sun are:

$$a_{\min} = 4.95 \times 10^{10} \text{ m} \quad \text{and} \quad a_{\max} = 6.16 \times 10^{11} \text{ m}$$

Calculate the orbital period $t_{\text{Encke}}$ of Comet Encke. Express the final answer in SI units.

---

| △ **BASE MODEL** | ✓ **RL-FINETUNED MODEL** |
|---|---|

**△ BASE MODEL**

**Step 1:** Formula
Use Kepler's Third Law:
$T^2 = (4\pi^2/GM) \cdot a^3$

**Step 2:** Calculate Semi-major Axis
$a = (a_{\min} + a_{\max})/2$
$a = (4.95 \times 10^{10} + 6.16 \times 10^{11})/2$
$a = 33.275 \times 10^{10}$ m

**Step 3:** Calculate Period
Using $\mu = 1.327 \times 10^{20}$:
$T^2 = (33.275 \times 10^{10})^3/1.327 \times 10^{20}$
$T^2 = 3.7074 \times 10^{33}/1.327 \times 10^{20}$

> **ERROR**: Arithmetic. $a^3 \approx 3.7 \times 10^{34}$
> (Order of magnitude off).

$T^2 \approx 2.794 \times 10^{13}$ s$^2$
$T \approx 5.285 \times 10^6$ s
$T \approx 61.37$ days
**Final Answer:**

$\boxed{61.37 \text{ days}}$ **(Incorrect)**

**✓ RL-FINETUNED MODEL**

**Step 1:** Calculate Semi-major Axis
$a = (a_{\min} + a_{\max})/2$
$a = (4.95 \times 10^{10} + 61.6 \times 10^{10})/2$
$a = 3.3275 \times 10^{11}$ m

**Step 2:** Unit Conversion Strategy
Convert $a$ to Astronomical Units (AU) to simplify Kepler's Law ($T^2 = a^3$ in years/AU).
1 AU = $1.496 \times 10^{11}$ m
$a_{au} = 3.3275 \times 10^{11}/1.496 \times 10^{11} \approx 2.225$ AU

**Step 3:** Apply Kepler's Law
$T^2 = (2.225)^3 \approx 10.96$
$T \approx \sqrt{10.96} \approx 3.31$ years

**Step 4:** Convert to SI
1 year = $3.154 \times 10^7$ seconds
$T = 3.31 \times 3.154 \times 10^7$
$T \approx 1.04 \times 10^8$ seconds
**Final Answer:**

$\boxed{1.04 \times 10^8 \text{ s}}$ **(Correct)**

*Figure 29.* **LLM answers before (left) and after (right) RL finetuning.** Question adapted from IPhO 2013 Question 1 "The Maribo Meteorite".

### F=MA 2024 Q17

Consider the following system of massless and frictionless pulleys, ropes, and springs. Initially, a block of mass $m$ is attached to the end of a rope, and the system is in equilibrium. Next the block is doubled in mass, and the system is allowed to come to equilibrium again. During the transition between these equilibria, how far does the end of the rope (where the block is suspended) move?

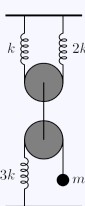

---

### INVENTED ENTITIES AND NATURAL-LANGUAGE DESCRIPTIONS

**Entity 1:** `3-point-movable-pulley`

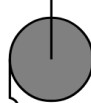

A movable pulley of mass `<mass:float>` kg is used as a reusable transmission node. It exposes independent left and right branches and a center hook on the `<side:up/down>` side.

**Entity 2:** `anchored spring`

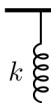

A spring with stiffness `<k:float>` N/m and natural length `<length:float>` m is anchored to a fixed support. The spring points toward `<dir:up/down/left/right>` and its movable endpoint mass is `<m:float>` kg. The same endpoint is exposed for connection to an external light string.

---

### △ DIRECT XML ATTEMPT

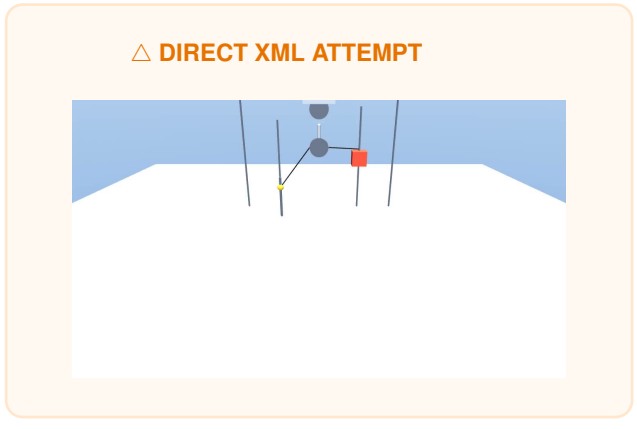

### ✓DSL-GENERATED SIMULATION

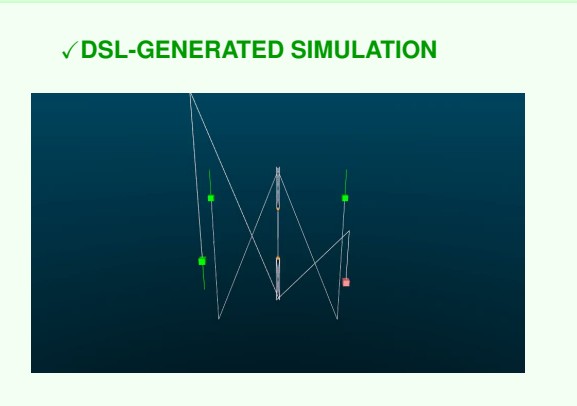

*Figure 30.* Scaling up the DSL using LLMs. In this experiment, we task the LLM to extend our entity vocabulary by inventing new entities to support simulation of this question from **F=MA 2024**. We observe that while modern LLMs fail to simulate a target scene by generating raw simulator code (left), they can do so by extending our DSL with novel entities (right).

## USA PHO 2019 B3

A bead of mass $M$ slides frictionlessly along a horizontal rail. It is attached to a rigid, massless rod of length $R$ with a ball of mass $M$ at the other end. The system is initially stationary with the ball directly above the bead ($\theta = 0$) before receiving an infinitesimal horizontal push. The rail only constrains the bead, allowing the rod and ball to pass through unhindered.

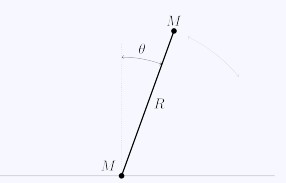

### INVENTED ENTITIES AND NATURAL-LANGUAGE DESCRIPTIONS

**Entity 1:** `slider mass`

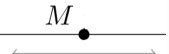

A carriage of mass `<mass:float>` kg is constrained to horizontal translation. The carriage has an exposed top pivot and side connectors for attaching strings.

**Entity 2:** `light rod pendulum`

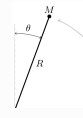

A rigid rod of length `<length:float>` m carries a bob of mass `<mass:float>` kg. The assembly rotates about a top pivot, initially tilted at an angle of `<slope:float>` degrees from horizontal.

✓**DIRECT XML ATTEMPT**

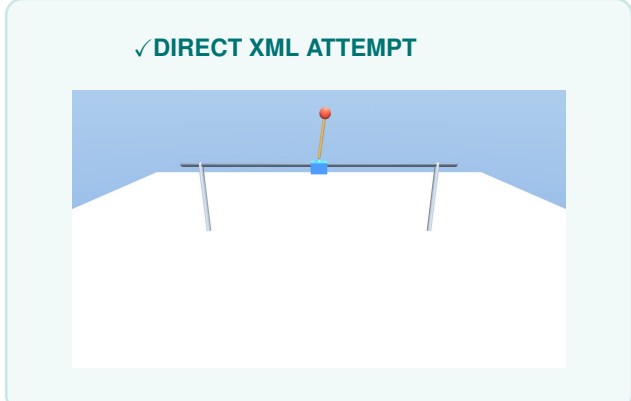

✓**DSL-GENERATED SIMULATION**

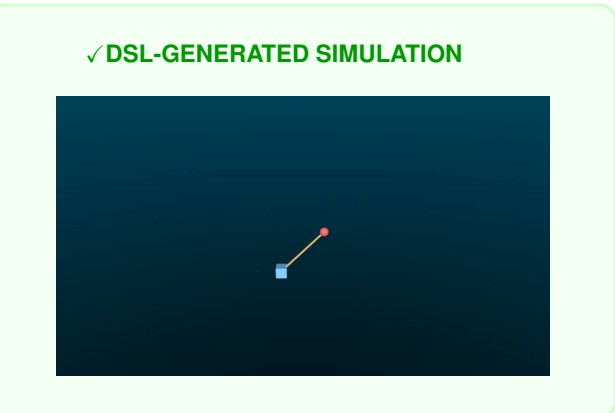

*Figure 31.* Scaling up the DSL using LLMs. In this experiment, we task the LLM to extend our entity vocabulary by inventing new entities to support simulation of this question from **USA PhO 2019**. In this case, we observe that LLMs are able to simulate a target scene by generating raw simulator code (left), and by extending our DSL with novel entities (right).

## JEE ADVANCED 2019 PAPER 2

A block of mass $2M$ is attached to a massless spring with spring constant $k$. This block is connected to two other blocks of masses $M$ and $2M$ using two massless pulleys and strings. The accelerations of the blocks are $a_1$, $a_2$, and $a_3$ as shown in the figure. The system is released from rest with the spring in its unstretched state. The maximum extension of the spring is $x_0$. Which of the following option(s) is/are correct? [$g$ is the acceleration due to gravity. Neglect friction.]

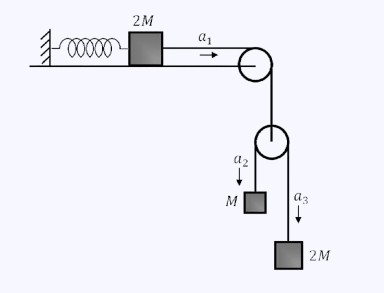

### INVENTED ENTITIES AND NATURAL-LANGUAGE DESCRIPTIONS

**Entity 1:** `spring mass`

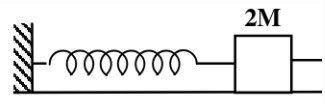

A block of mass `<mass:float>` kg can slide on a plane inclined at `<angle:float>` degrees. Its left side is attached to a spring fixed to a wall. The spring has stiffness `<k:float>` N/m and natural length `<length:float>` m. The right side of the block is available for connection to another entity by a light string.

### △ DIRECT XML ATTEMPT

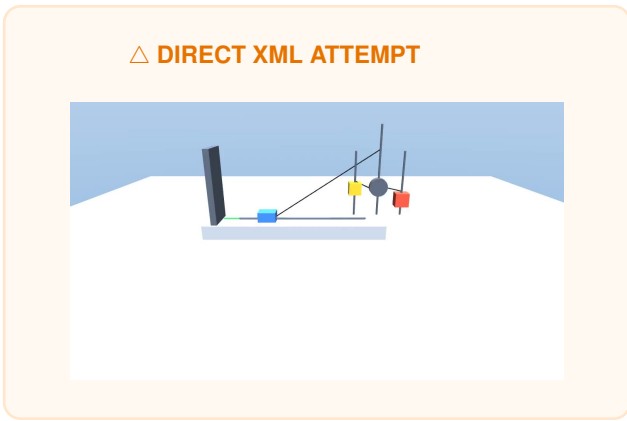

### ✓DSL-GENERATED SIMULATION

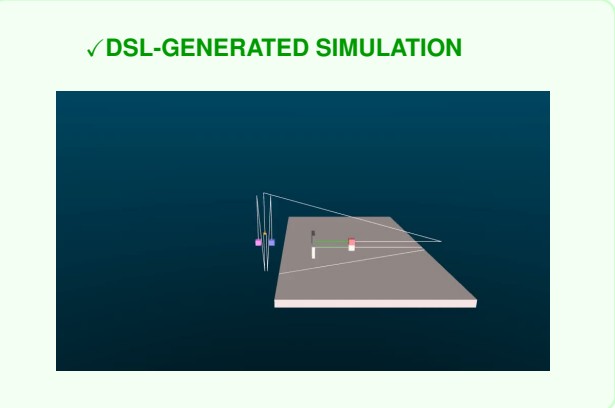

*Figure 32.* Scaling up the DSL using LLMs. In this experiment, we task the LLM to extend our entity vocabulary by inventing new entities to support simulation of this question from **JEE Advanced 2019**. We observe that while LLMs fail to simulate a target scene by generating raw simulator code (left), they can do so by extending our DSL with novel entities (right).

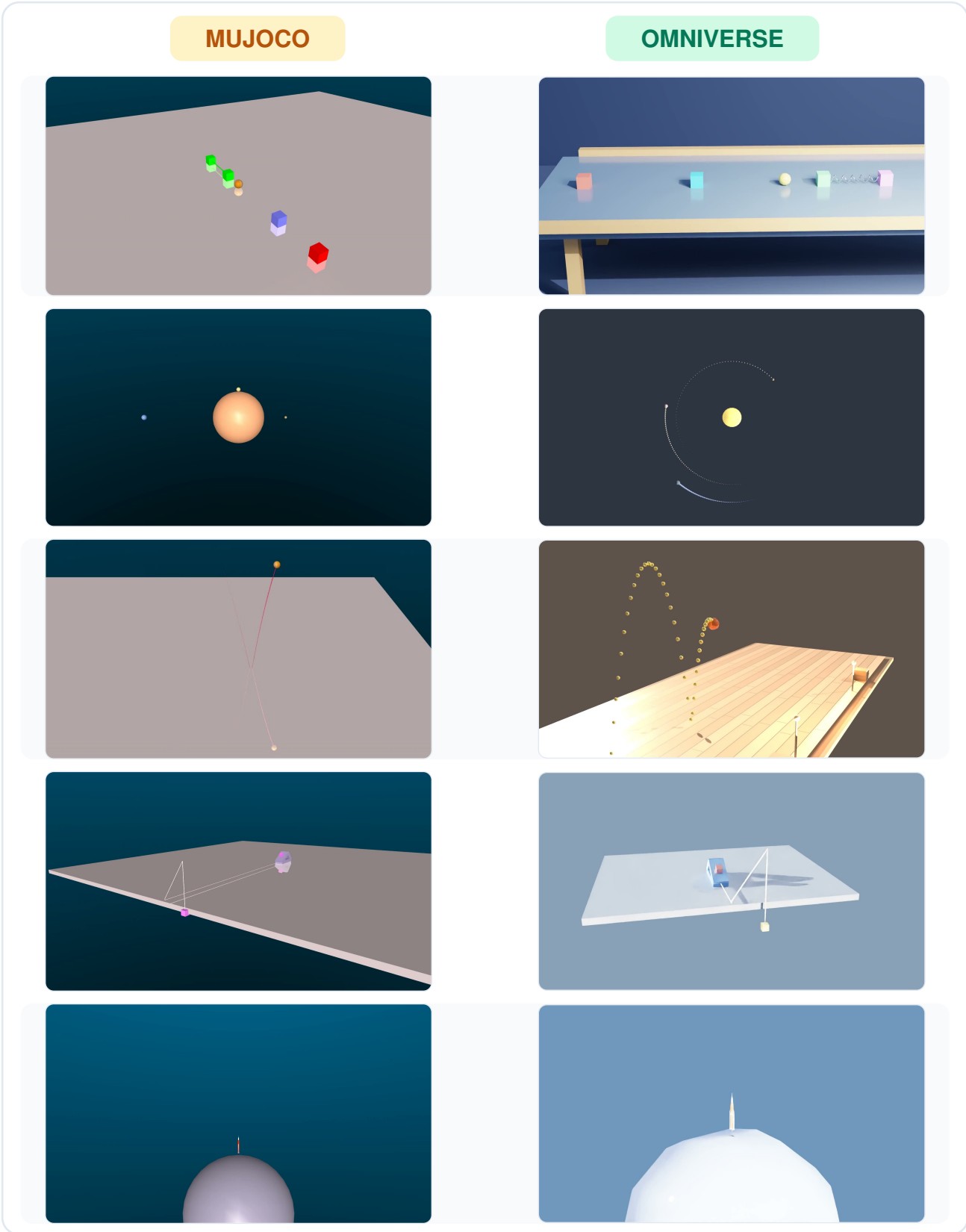

*Figure 33.* We task LLMs to port a subset of our entities from MuJoCo simulator to Omniverse simulator, to evaluate the portability of our DSL across simulators. LLMs could successfully transfer DSL entities across simulators, which could potentially help expand the scope of simulator-based synthetic data generation.

