# OpenReview forum: "Sim2Reason: Solving Physics Olympiad via Reinforcement Learning on Physics Simulators"
_ICML.cc/2026/Conference — ICML 2026 regular_

### Official Review · Reviewer_wBCg · 2026-03-08

**Soundness:** 3
**Presentation:** 3
**Significance:** 3
**Originality:** 2
**Overall Recommendation:** 4
**Confidence:** 3

**Summary:**

This paper proposes to use physics simulators as a source of making synthetic data. The authors design a pipeline to create synthetic physics data based on a predefined design language and set of objects. The pipeline uses MoJuCo to simulate a scenario and extract problem answer pairs. These problem-answer pairs are then used to perform RL on LLMs and the authors show performance improvements on the international physics olympiad as a results of this RL training.

**Compliance With Llm Reviewing Policy:**

Affirmed.

**Final Justification:**

My concerns are mainly addressed and I raised my score to 4

**Key Questions For Authors:**

* In section 2.5, you mention "with a reference policy $\pi_{ref}$ (the base Instruct model).", then in equation (2) you use the $\pi_{ref}$ as the old model. Is that a typo? There should be a distinction between the reference model and the old policy model (from which the data was generated).

**Limitations:**

yes

**Strengths And Weaknesses:**

Strengths:
* Using synthetic data to improve performance on IPhO is interesting.
* The filtering mechanism to filter out the data where some part of the scene is ignored is very clever!
* The experimental results show performance improvement over the baseline.

Weaknesses:
* The related work is very thin. There has been a lot of work in terms of using synthetic data to improve performance on real benchmarks. For instance, AlphaGeometry uses synthetic geometry problem generators using a symbolic engine to improve performance on real geometry olympiad problems.
* There is very little discussion of how how the domain specific language is made, and there seems to be a lot of manual work to incorporate interesting set of objects and entities into the system. Therefore, I cannot evaluate how much it is worth it to extend this system, compared to preparing templates of physics questions and just changing the numbers (similar to the GSM-Symbolic work). I would encourage the authors to elaborate more on the scene generation and whether LLMs have been used there (or whether LLMs could facilitate scaling that up)
* In Figure 8, it seems that this dataset is harder than IPhO for LLMs (e.g., GPT-5's performance on IPhO is around 41% while on the authors' dataset is less than 13%). Which brings the question whether this dataset is suitable for RL (as the RL advantages would be near zero for most models). Is the difficulty tunable here? In figure 4, I see the Qwen3-30B performance on the validation set starts with 17%, which suggests it is stronger than GPT5 on this dataset?

Overall, I believe the paper makes an interesting observation that using physical simulators could help with real problems, however, I am not a physics expert, and not sure how much manual work is done curating interesting objects and their relations, vs the simulator doing the heavy work.

---

> ### Author Rebuttal · Authors · 2026-03-31
>
> We thank the reviewer for appreciating our approach of using synthetic data for improving in IPhO and the shortcut filtering mechanism.
>
> - **Q4.1: More Baselines trained on Real World Scientific QA Data.** (general concern of reviewers)
>
> To provide a stronger comparison, we evaluated our approach against recent open-weight post-trained models of similar scale trained on real world QA data. The most important comparison is with **P1: Mastering Physics Olympiads with Reinforcement Learning** (arxiv:2511.13612), which is post-trained on over 5000 QA pairs curated from physics olympiads and various textbooks. Additionally we compare against DAPO-32B and LIMO-32B which are trained on math qa dataset. We find that our model, trained entirely on synthetic simulator-generated data, outperforms these baselines.
>
> | Base Model | Post-Trained Model | IPhO Accuracy (%) |
> |---|---|---:|
> | Qwen 2.5 (32B) | DAPO 32B | 24.7 |
> | Qwen 2.5 (32B) | LIMO 32B | 25.5 |
> | Qwen 3 (30B) | Prime P1 30B | 38.6 |
> | Qwen 3 (30B) | **Ours (Sim2Reason) 30B** | **40.0** |
>
> We will add this table and broaden related work accordingly.
>
>
> - **Q4.2: Creating of the DSL and scaling it up with LLMs.**
>
> We design entities by capturing recurring physical motifs that appear frequently in high-school physics textbooks. When introducing a new entity, we intentionally keep it general, reusable, and compositional, rather than making it specific to a single question. This allows to gneerate exponentially many unique scenes. We find that our synthetic data results in improvemnets across a braod range of physics benchmarks (IPhO, JEEBench, OlympiadBench, PHYSICS), with difficulties ranging from high school olympiad to undergraduate level, including questions that cannot be simulated with our DSL or the simulator. Additionally, we conduct experiments to test the scalability of our approach and find that **LLMs can extend our DSL by inventing new entities to simulate previously unsupported scenes**.
>
> We conducted the following experiment:
>
> 1. Retrieval. A GPT-5.4 agent parsed real-world exam sources (F=ma, USAPhO, and JEE Advanced) and identified 3 mechanics questions that could not be expressed with our current DSL library.
> 2. Generation (Direct XML vs. DSL). We then asked a GPT-5.3 Codex (high) agent to simulate these questions, in MuJoCo using two routes:
>    - Raw MuJoCo XML, and
>    - Our DSL space.
> 3. Results.
>    - RAW XML generation: Had a 33% success rate (1 out of 3 target questions simulated). Failures were typically due to incorrect spatial referencing, joint setup, rope routing or missing bodies.
>    - DSL entity generation: Had a 100% success rate (3 out of 3 target questions simulated). With the DSL abstractions in place, the agent successfully wrote new reusable entities with only minor hints/corrections after visual inspection (e.g., axis alignment or connection orientation).
>
> **Takeaway.** Raw simulator code is too unstructured to scale reliably with current LLMs. Our DSL provides the right abstraction layer for LLMs to design reusable entities, therefore substantially reduces the manual curation bottleneck and makes the pipeline scalable.
>
> We provide supplementary materials including: **(1)** the source benchmark questions, **(2)** the designed entities and their natural language description templates, **(3)** videos of successful MuJoCo simulations, and **(4)** examples of unsuccessful direct-XML attempts. These materials can be found at [https://physics-rl.github.io/rebuttal_exp1/](https://physics-rl.github.io/rebuttal_exp1/).
>
>
> - **Q4.3: Figure 4 vs. Figure 8 discrepancy.**
>
> Figure 4 shows a small online validation subset used during training for fast monitoring, whereas Figure 8 reports the full evaluation set. To prevent confusion in the future, we will add the base Qwen3-30B to the plot in Figure 8 to help readers better understand the standings of various models. We add the Qwen 3 30B alongside rest of the models in Figure 8, in a table below to address the concern by the reviewer. Qwen 3 30B does not have a particularly high performance than other reasoning models such as GPT-5 high as seen in the table.
>
> | Model | Synthetic Accuracy (%) | IPhO Accuracy (%) |
> |---|---:|---:|
> | Claude Sonnet 4.5 | 7.0 | 29.9 |
> | GPT 5 Nano | 7.2 | 35.0 |
> | GPT 5 Mini | 8.5 | 36.4 |
> | Gemini 2.5 Flash | 8.9 | 41.6 |
> | Qwen 3 30B Instruct | 10.6 | 35.6 |
> | GPT 5 High | 12.9 | 41.6 |
>
> - **Q4.4: Expanding related work.**
>
> Thank you for the suggestion. We will include relevant discussion of AlphaGeometry, Rubik's Cube with a Robot Hand, MetaMath, and Self-Instruct in the related work. While these works use procedurally generated synthetic data for training, Sim2Reason generates structured QA data from physics simulators, moving beyond tasks such as math/code.
>
> - **Q4.5: Notation in Sec. 2.5 / Eq. (2).**
>
> Yes, this is a typo—$\pi_{old}$ should replace $\pi_{ref}$ in Eq. (2). We'll fix this in the final version.

---

> > ### Author Rebuttal · Reviewer_wBCg · 2026-04-01
> >
> > Thank you for the rebuttal. My concerns are mainly addressed and I raise my score.

---

### Official Review · Reviewer_uVqJ · 2026-03-11

**Soundness:** 3
**Presentation:** 3
**Significance:** 2
**Originality:** 3
**Overall Recommendation:** 4
**Confidence:** 4

**Summary:**

This paper proposes Sim2Reason, a framework that uses physics simulators to generate synthetic question–answer pairs for training language models on physics reasoning. The method procedurally generates physical scenes, simulates their dynamics, converts the simulation traces into natural language questions, and then trains the model using reinforcement learning with verifiable rewards. The authors also introduce a shortcut filtering step that removes questions where simplified reasoning leads to the same answer. Experiments on several physics benchmarks show consistent improvements after training on the generated data. The paper explores whether physics simulators can serve as scalable sources of supervision for training reasoning-capable language models, addressing the lack of large-scale QA data in scientific domains.

**Compliance With Llm Reviewing Policy:**

Affirmed.

**Final Justification:**

Thank the authors for their helpful clarifications. My main concerns have been addressed, so I raise my score to 4.

**Key Questions For Authors:**

1. What fraction of generated QA pairs are removed by shortcut filtering?
2. Can you provide some examples where shortcut solution actually happens?

**Limitations:**

yes

**Strengths And Weaknesses:**

**Strength**

1. The paper presents a compelling approach to using physics simulators as large-scale data generators for reasoning tasks, which could be useful for domains where labeled datasets are scarce.

2. The experiments show improvements across several evaluation datasets, suggesting that synthetic simulator data can transfer to real-world physics problems.

3. The proposed filtering mechanism attempts to address reward hacking issues by removing questions that can be solved by ignoring parts of the physical system.

**Weakness**

1. Lack of statistics on shortcut filtering.
The paper motivates shortcut filtering but does not provide quantitative statistics on how frequently shortcut-solvable questions occur in the generated dataset. It would be helpful to report the proportion of generated QA pairs removed by filtering and how often entity or joint removal changes the answer.

2. Limited baselines.
The experimental baselines are relatively limited, as the evaluation mainly focuses on Qwen models and compares performance before and after RL training. It would strengthen the paper to include additional models as baselines. For example, comparing with other models of similar size, or models that have been fine-tuned specifically for physics or scientific reasoning, would provide a clearer picture of the effectiveness of the proposed approach.

3. Dependence on initial model capability.
RL with outcome-only rewards relies on the base model already having non-trivial success probability. If the model cannot solve a class of problems at all, RLVR may struggle to discover correct reasoning trajectories due to sparse rewards. The paper does not analyze how sensitive the approach is to the initial model capability.

4. Domain specificity of the approach.
The proposed filtering and data generation rely heavily on the ability to perform counterfactual simulations. While this works for physics domains with accurate simulators, it is unclear how the method would generalize to reasoning tasks without simulators (e.g., mathematics or symbolic reasoning).

---

> ### Author Rebuttal · Authors · 2026-03-31
>
> We thank the reviewer for acknowledging the novelty and extensive experimentation in our work. We address their individual concerns along with a general concern raised by the reviewers.
>
> - **Q3.1: Shortcut filtering.**
>
> In general, we observe that roughly **15%** of the generated questions are filtered for having a shortcut solution.  In the paper, we have provided an example of a scene which can have shortcut solution. We provide a few example solutions for that question by Gemini 3.1 Pro where the model approximates the solution by taking a shortcut at this link: [https://physics-rl.github.io/rebuttal_exp3/](https://physics-rl.github.io/rebuttal_exp3/).
>
> > It would be helpful to report the proportion of generated QA pairs removed by filtering and how often entity or joint removal changes the answer.
>
> We believe there may have been a small misunderstanding regarding shortcut-filtering: we do **not** remove entities or joints from the final training set itself. Instead, we generate simplified counterfactual scenes and remove the original **question** if the queried answer is unchanged. A single scene can therefore yield both valid and shortcut-solvable questions; only the latter are filtered out.
>
> - **Q3.2: More Baselines trained on Real World Scientific QA Data.**
>
> To provide a stronger comparison, we evaluated our approach against recent open-weight post-trained models of similar scale trained on real world QA data. The most important comparison is with **P1: Mastering Physics Olympiads with Reinforcement Learning** (arxiv:2511.13612), which is post-trained on over 5000 QA pairs curated from physics olympiads and various textbooks. Additionally we compare against DAPO-32B and LIMO-32B which are trained on math qa dataset. We find that our model, trained entirely on synthetic simulator-generated data, outperforms these baselines.
>
> | Base Model | Post-Trained Model | IPhO Accuracy (%) |
> |---|---|---:|
> | Qwen 2.5 (32B) | DAPO 32B | 24.7 |
> | Qwen 2.5 (32B) | LIMO 32B | 25.5 |
> | Qwen 3 (30B) | Prime P1 30B | 38.6 |
> | Qwen 3 (30B) | **Ours (Sim2Reason) 30B** | **40.0** |
>
> We will add this table and broaden related work accordingly.
>
> - **Q3.3: Dependence on initial model capability.**
>
> In this work, we observe consistent gains across model scales and capabilities - which ranges from weak models such as Qwen 2.5 3B to strong models such as Qwen3-30B - suggesting that the gains by our approach does not rely on base model capabilities. Additionally, the synthetic data generation naturally supports curriculum tuning to improve pass@k, since our dataset is generated procedurally in simulation,  the difficulty can be controlled by the number of entities, coupling depth, and presence or absence of physical phenomena (such as friction, collision, etc).
>
> - **Q3.4: Generalization to reasoning tasks without simulators (e.g., mathematics or symbolic reasoning).**
>
> While remarkable progress has been witnessed in LLMs solving Math Olympiads (such as Winning Gold at IMO 2025 with a Model-Agnostic Verification-and-Refinement Pipeline, arxiv:2507.15855), performance of LLMs in sciences (such as Physics) lags behind, owing to the scarcity of scientific post-training data. Therefore, in this work we primarily focus on improving physical reasoning of LLMs by generating synthetic data using simulators. Further, we find that training on our synthetic data also improves the model on math benchmarks such as AIME from 10.83% to 12.5% (Table 2 in the paper).
>
> - **Q3.5: Scaling Up the DSL with LLMs.** (general concern of reviewers)
>
> To test the scalability of our approach, we conduct additional experiments and find that **LLMs can extend our DSL by inventing new entities to simulate previously unsupported scenes**.
>
> For details on the steps of experiment, please refer to Q2.3.
>
> **Takeaway.** Raw simulator code is too unstructured to scale reliably with current LLMs. Our DSL provides the right abstraction layer for LLMs to design reusable entities, therefore substantially reduces the manual curation bottleneck and makes the pipeline scalable.
>
> We provide supplementary materials including: **(1)** the source benchmark questions, **(2)** the designed entities and their natural language description templates, **(3)** videos of successful MuJoCo simulations, and **(4)** examples of unsuccessful direct-XML attempts. These materials can be found at [https://physics-rl.github.io/rebuttal_exp1/](https://physics-rl.github.io/rebuttal_exp1/).

---

> > ### Author Rebuttal · Reviewer_uVqJ · 2026-04-03
> >
> > Thank the authors for their helpful clarifications. My main concerns have been addressed, so I raise my score to 4.

---

### Official Review · Reviewer_Cuyz · 2026-03-13

**Soundness:** 4
**Presentation:** 4
**Significance:** 4
**Originality:** 4
**Overall Recommendation:** 5
**Confidence:** 4

**Summary:**

This paper presents SIM2REASON, a framework that addresses the data scarcity bottleneck in physical reasoning for LLMs by utilizing physics engines as scalable, high-fidelity data generators. The authors develop a Domain-Specific Language (DSL) to procedurally generate diverse physical scenarios in MuJoCo, creating verifiable question-answer pairs (numeric, symbolic, and reverse reasoning). By employing Reinforcement Learning with Verifiable Rewards (RLVR), the model internalizes physical principles from synthetic "executable textbooks" rather than static corpora. The work demonstrates strong zero-shot sim-to-real transfer performance on expert-level benchmarks, including the International Physics Olympiad (IPhO) and JEE-Bench.

**Compliance With Llm Reviewing Policy:**

Affirmed.

**Final Justification:**

The authors have largely addressed my concerns, and I will maintain my positive score.

**Key Questions For Authors:**

1.	Generalization: How portable is the DSL to domains like Electromagnetism or Fluid Dynamics where MuJoCo has inherent limitations?

**Limitations:**

yes

**Strengths And Weaknesses:**

### Strengths
•	Originality: The integration of a procedural DSL for scene generation with RLVR is a novel approach for physical reasoning. It effectively transforms the simulator into an infinite, self-correcting supervision signal.
•	Significance: This work has high potential impact by providing a blueprint for scaling scientific LLMs in domains where human-labeled data is expensive or scarce.
•	Soundness: The evaluation is comprehensive, testing across multiple model scales (3B to 72B) and diverse real-world benchmarks, validating generalized physical understanding.

### Weaknesses
•	Simulator Bias: The reliance on the MuJoCo engine may introduce specific modeling biases. If the simulator's approximation of physical laws is slightly inaccurate, the model might internalize these artifacts as ground truth.

---

> ### Author Rebuttal · Authors · 2026-03-31
>
> We thank the reviewer for recognizing the potential impact of the work and the novelty of procedural scene generation for synthetic data generation. We address their individual concerns along with two general concerns raised by other reviewers.
>
> - **Q2.1: Simulator bias.**
>
> Our reward function partially accounts for simulator inaccuracy by using a tolerance of $5\times 10^{-2}$: if the model answer lies within 5% relative error of the simulator-recorded value, it receives positive reward. We chose this threshold empirically by inspecting model responses on a small validation set. In practice, responses that use the correct underlying physics but differ slightly due to numerical approximation or formatting almost always fall within this band, whereas responses based on incorrect physical reasoning tend to lie well outside it.
>
> - **Q2.2: DSL portability.**
>
> A key advantage of our approach is that scene generation happens at a high level of abstraction, namely entities and connections, rather than raw simulator syntax. As a result, porting to a different simulator for domains such as fluid dynamics or electromagnetism would mainly require re-implementing the entities and connections in the new backend. Modern LLMs are increasingly capable of assisting with this process. **To test this directly, we ran an additional experiment in which LLMs ported a subset of our MuJoCo entities to NVIDIA Omniverse.** For all entities supported by Omniverse's physics engine, the transfer was successful. We provide example scenes in MuJoCo and in Omniverse here: [https://physics-rl.github.io/rebuttal_exp2/](https://physics-rl.github.io/rebuttal_exp2/). Together with the DSL-extension experiment described in Q2.3, this suggests that both extending the vocabulary and porting it across simulators with the help of LLMs, is quite practical.
>
> - **Q2.3: Scaling Up the DSL with LLMs.** (general concern of reviewers)
>
> To test the scalability of our approach, we conduct additional experiments and find that **LLMs can extend our DSL by inventing new entities to simulate previously unsupported scenes**.
>
> We conducted the following experiment:
> 1. Retrieval. A GPT-5.4 agent parsed real-world exam sources (F=ma, USAPhO, and JEE Advanced) and identified 3 mechanics questions that could not be expressed with our current DSL library.
> 2. Generation (Direct XML vs. DSL). We then asked a GPT-5.3 Codex (high) agent to simulate these questions, in MuJoCo using two routes:
>    - Raw MuJoCo XML, and
>    - Our DSL space.
> 3. Results.
>    - RAW XML generation: Had a 33% success rate (1 out of 3 target questions simulated). Failures were typically due to incorrect spatial referencing, joint setup, rope routing or missing bodies.
>    - DSL entity generation: Had a 100% success rate (3 out of 3 target questions simulated). With the DSL abstractions in place, the agent successfully wrote new reusable entities with only minor hints/corrections after visual inspection (e.g., axis alignment or connection orientation).
>
> **Takeaway.** Raw simulator code is too unstructured to scale reliably with current LLMs. Our DSL provides the right abstraction layer for LLMs to design reusable entities, therefore substantially reduces the manual curation bottleneck and makes the pipeline scalable.
>
> We provide supplementary materials including: **(1)** the source benchmark questions, **(2)** the designed entities and their natural language description templates, **(3)** videos of successful MuJoCo simulations, and **(4)** examples of unsuccessful direct-XML attempts. These materials can be found at [https://physics-rl.github.io/rebuttal_exp1/](https://physics-rl.github.io/rebuttal_exp1/).
>
> - **Q2.4: More Baselines trained on Real World QA Data.** (general concern of reviewers)
>
> To provide a stronger comparison, we evaluated our approach against recent open-weight post-trained models of similar scale trained on real world QA data. The most important comparison is with **P1: Mastering Physics Olympiads with Reinforcement Learning** (arxiv:2511.13612), which is post-trained on over 5000 QA pairs curated from physics olympiads and various textbooks. Additionally we compare against DAPO-32B and LIMO-32B which are trained on math qa dataset. We find that our model, trained entirely on synthetic simulator-generated data, outperforms these baselines.
>
> | Base Model | Post-Trained Model | IPhO Accuracy (%) |
> |---|---|---:|
> | Qwen 2.5 (32B) | DAPO 32B | 24.7 |
> | Qwen 2.5 (32B) | LIMO 32B | 25.5 |
> | Qwen 3 (30B) | Prime P1 30B | 38.6 |
> | Qwen 3 (30B) | **Ours (Sim2Reason) 30B** | **40.0** |
>
> We will add this table and broaden related work accordingly.

---

> > ### Author Rebuttal · Reviewer_Cuyz · 2026-04-03
> >
> > Thank you for the rebuttal. The authors have largely addressed my concerns, and I will maintain my positive score.

---

### Official Review · Reviewer_PEo6 · 2026-03-13

**Soundness:** 3
**Presentation:** 3
**Significance:** 3
**Originality:** 2
**Overall Recommendation:** 4
**Confidence:** 4

**Summary:**

The paper proposes Sim2Reason, a framework that uses physics simulators to automatically generate large amounts of physics question–answer data for training language models. Instead of relying on limited internet QA pairs, the authors train LLMs with reinforcement learning on simulator-generated problems. The results show that models trained this way improve their performance on real-world physics benchmarks like the International Physics Olympiad.

**Compliance With Llm Reviewing Policy:**

Affirmed.

**Final Justification:**

I maintain my positive score and decision.

**Key Questions For Authors:**

See weaknesses

**Limitations:**

See weaknesses

**Strengths And Weaknesses:**

Strengths:
1The proposed method shows a certain degree of novelty. It leverages physics simulators to generate training supervision signals and converts simulation trajectories into multiple QA formats for reinforcement learning.
2.The data generation framework is scalable. It can automatically generate a large variety of physical systems, enabling large-scale data generation and alleviating the scarcity of real-world data.
3.The experimental design is relatively comprehensive.

Weakness:

The paper trains exclusively on synthetic data and lacks comparisons with existing works that also leverage synthetic or curated physics problems for reinforcement learning. This makes it difficult to assess whether the proposed data generation approach offers a tangible advantage over prior methods. In particular, recent studies such as P1: Mastering Physics Olympiads with Reinforcement Learning (arXiv:2511.13612) and Enhancing LLMs for Physics Problem-Solving using Reinforcement Learning with Human-AI Feedback (arXiv:2412.06827) provide relevant baselines. Including comparisons with these approaches would better demonstrate the effectiveness of the proposed method.

---

> ### Author Rebuttal · Authors · 2026-03-31
>
> We thank the reviewer for their thoughtful feedback. We address their individual concern along with a general concern raised by reviewers.
>
> - **Q1.1: Comparision against P1 and other baselines trained on Real World Data**
>
> To provide a strong comparision, we evaluated our approach against recent open-weight post-trained models of similar scale, trained on real world QA datasets. The most important comparision is with **P1: Mastering Physics Olympiads with Reinforcement Learning** (arxiv:2511.13612), which is post-trained on over 5000 QA pairs curated from physics olympiads and various textbooks. Additionally we compare our model against DAPO-32B and LIMO-32B, which are trained on math QA datasets. We find that our model **trained entirely on synthetic simulator-generated data** outperforms these baselines.
>
> | Base Model | Post-Trained Model | IPhO Accuracy (%) |
> |---|---|---:|
> | Qwen 2.5 (32B) | DAPO 32B | 24.7 |
> | Qwen 2.5 (32B) | LIMO 32B | 25.5 |
> | Qwen 3 (30B) | Prime P1 30B | 38.6 |
> | Qwen 3 (30B) | **Ours (Sim2Reason) 30B** | **40.0** |
>
> We couldn't compare our model against Enhancing LLMs for Physics Problem-Solving using Reinforcement Learning with Human-AI Feedback (arXiv:2412.06827) as their model weights are not open sourced. We thank the reviewer for pointing out these baselines. We will add this table and broaden related work accordingly.
>
> - **Q1.2: Scaling Up the DSL with LLMs** (general concern of reviewers)
>
> To test the scalability of our approach, we conduct additional experiments and find that **LLMs can extend our DSL library by inventing new entities to simulate previously unsupported scenes.**  We conducted the following experiments:
>
> 1. **Retrieval.** A GPT-5.4 agent parsed real-world exam sources (F=ma, USAPhO, and JEE Advanced) and identified 3 mechanics questions that could not be expressed with our current DSL library.
> 2. **Generation (Direct XML vs. DSL).** We then asked a GPT-5.3 Codex (high) agent to simulate these questions, in MuJoCo using two routes:
>    - **Raw MuJoCo XML**, and
>    - **Our DSL space.**
> 3. **Results.**
>    - **RAW XML generation:** Had a **33% success rate** (1 out of 3 target questions simulated). Failures were typically due to incorrect spatial referencing, joint setup, rope routing or missing bodies.
>    - **DSL entity generation:** Had a **100% success rate** (3 out of 3 target questions simulated). With the DSL abstractions in place, the agent successfully wrote new reusable entities with only minor hints/corrections after visual inspection (e.g., axis alignment or connection orientation).
>
> **Takeaway.** Raw simulator code is too unstructured to scale reliably with current LLMs. Our DSL provides the right abstraction layer for LLMs to design reusable entities, therefore substantially reduces the manual curation bottleneck and makes the pipeline scalable.
>
> We provide supplementary materials including: **(1)** the source benchmark questions, **(2)** the designed entities and their natural language description templates, **(3)** videos of successful MuJoCo simulations, and **(4)** examples of unsuccessful direct-XML attempts. These materials can be found at [https://physics-rl.github.io/rebuttal_exp1/](https://physics-rl.github.io/rebuttal_exp1/).

---

> > ### Author Rebuttal · Reviewer_PEo6 · 2026-04-04
> >
> > The authors have largely addressed my concerns, and I will maintain my positive score.

---

### Decision · Program_Chairs · 2026-04-30

**Decision:**

Accept (regular)

**Comment:**

This paper proposes a framework called Sim2Reason, that generates physics question-answer data using physics simulators for post-training LLMs. The authors show that training on the generated data improves performance on International Physics Olympiad problems. Overall, this work shows that physics simulators can be useful as synthetic data generators that allow LLMs to learn physical reasoning skills.

Reviewer PEo6 found the proposed method to be novel, the data generation framework to be scalable, and the experimental evaluation to be comprehensive.

Reviewer Cuyz found the paper to be novel and high-impact, and found the evaluation to be comprehensive.

Reviewer uVqJ found the approach proposed in the paper to be compelling. They found that the experiments show improvements across multiple datasets.

Reviewer wBCg found the idea to be interesting, the filtering mechanism to be clever, and the experimental results to be strong.

Reviewer PEo6 raised concerns regarding how the framework trains only on synthetic data, and how the paper lacks comparisons with prior works that use synthetic or curated physics problems for RL post-training. The reviewer asked for comparisons to two baselines, “Mastering Physics Olympiads with RL” and “Enhancing LLMs for Physics Problem-Solving using Reinforcement Learning with Human-AI Feedback.”

In their rebuttal, the authors provided a comparison to one of the baselines the reviewer suggested (the authors left the other baseline out because it did not have open-source weights).

Reviewer Cuyz raised a concern regarding simulator bias arising from the use of MuJoCo as the simulation engine. The reviewer also asked whether the framework can be extended to other domains like electromagnetism or fluid dynamics.

The authors’ rebuttal addressed all of the reviewer’s concerns.

Reviewer uVqJ raised concerns regarding the limited baselines used in the empirical comparison, only measuring the difference before and after post-training of Qwen models rather than including other baseline models that have been finetuned for physics tasks. The reviewer also raised concerns regarding the lack of a quantitative analysis of the effect of shortcut filtering. Finally, the reviewer raised a concern regarding the domain specificity of the approach and whether it could generalize to other types of reasoning, like math.

The rebuttal addressed all the reviewer’s concerns, and the reviewer raised their score.

Reviewer wBCg raised concerns regarding insufficient discussion of related work, and insufficient discussion of how the domain-specific language works.

The rebuttal addressed the reviewer’s concerns and the reviewer raised their score.

Overall, the reviewers were unanimous in their support for accepting this paper. All reviewers’ concerns were fully resolved by the rebuttals, and two reviewers increased their scores following the rebuttal period. This paper introduces an interesting and impactful approach to synthetic data generation for post-training to improve physics capabilities, and it will be of interest to the ICML community.